# Regret Matching$^+$: (In)Stability and Fast Convergence in Games

**Gabriele Farina**
MIT
gfarina@mit.edu

**Julien Grand-Clément**
ISOM, HEC Paris
grand-clement@hec.fr

**Christian Kroer**
IEOR, Columbia University
christian.kroer@columbia.edu

**Chung-Wei Lee**
Department of Computer Science
University of Southern California
leechung@usc.edu

**Haipeng Luo**
Department of Computer Science
University of Southern California
haipengl@usc.edu

## Abstract

Regret Matching$^+$ (RM$^+$) and its variants are important algorithms for solving large-scale games [35]. However, a theoretical understanding of their success in practice is still a mystery. Moreover, recent advances [34] on fast convergence in games are limited to no-regret algorithms such as online mirror descent, which satisfy *stability*. In this paper, we first give counterexamples showing that RM$^+$ and its predictive version [12] can be unstable, which might cause other players to suffer large regret. We then provide two fixes: restarting and chopping off the positive orthant that RM$^+$ operates in. Combined with RM$^+$ with predictions, we show that restarting is sufficient to get $O(T^{1/4})$ individual regret and that chopping off achieves $O(1)$ social regret in normal-form games. We also apply our stabilizing techniques to clairvoyant updates in the uncoupled learning setting for RM$^+$, introduced *Extragradient RM$^+$*, and prove desirable results akin to recent works for Clairvoyant online mirror descent [31, 14]. Our experiments show the advantages of our algorithms over vanilla RM$^+$-based algorithms in matrix and extensive-form games.

## 1 Introduction

Regret minimization is an important framework for solving games. Its connection to game theory provides a practically efficient way to approximate game-theoretic equilibria [16, 19]. Moreover, it provides a scaleable way to solve large-scale sequential games, for example using the *Counterfactual Regret Minimization* (CFR) decomposition [37]. Consequently, regret minimization algorithms are a central component in recent superhuman poker AIs [2, 28, 3]. *Regret Matching$^+$ (RM$^+$)* [35] is the most prevalent regret minimizer in these applications. In theory, it guarantees an $O(1/\sqrt{T})$ convergence rate after $T$ iterations, but its practical performance is usually significantly faster.

On the other hand, a line of recent works show that regret minimizers based on follow the regularized leader (FTRL) or online mirror descent (OMD) enjoy faster convergence rates in theory when combined with the concept of optimism/predictiveness [32, 34]. The result was originally proven

37th Conference on Neural Information Processing Systems (NeurIPS 2023).

| Algorithms | Social regret in multi-player NFGs |
| --- | --- |
| RM$^+$ [19] | $O\left(T^{1/2}\right)$ |
| Predictive RM$^+$ [10] | $O\left(T^{1/2}\right)$ |
| Stable Predictive RM$^+$ (Alg. 1) | $O\left(T^{1/4}\right)$ |
| Smooth Predictive RM$^+$ (Alg. 2) | $O(1)$ |
| Conceptual RM$^+$ (Alg. 3 ) | $O(1)$ |
| Approximate Conceptual RM$^+$ (Alg. 4 with $k = \log(T)$) | $O(1)$ |
| Extragradient RM$^+$ (Alg. 5) | $O(1)$ |

Table 1: Summary of regret guarantees for the algorithms studied in this paper. The constants hidden in the $O(\cdot)$ notations depends on initialization and the dimensions of the games and are given in our theorems.

in matrix games [32], and later extended to multiplayer normal-form games [34, 6, 7], extensive-form games [8, 10, 15, 1], and general convex games [22, 13]. However, despite their favorable properties in theory, optimistic algorithms based on FTRL/OMD are usually numerically inferior to RM$^+$ when applied to solving large-scale sequential games. It remains a mystery whether some optimistic variant of RM$^+$ enjoys a theoretically faster convergence rate, considering the strong empirical performance of RM$^+$. It is also an open question whether there exists an algorithm that has both favorable theoretical guarantees similar to FTRL/OMD algorithms and practical performance comparable to RM$^+$. Inspired by recent work on the connection between OMD and RM$^+$ [12], we provide new insights on the theoretical and empirical behavior of RM$^+$-based algorithms, and we show that the analysis of fast convergence for OMD can be extended to RM$^+$ with some simple modifications to the algorithm. Specifically, our main contributions can be summarized as follows.

1. We provide a detailed theoretical and empirical analysis of the potential for slow performance of RM$^+$ and predictive RM$^+$. We start by showing that, in stark contrast to FTRL/OMD algorithms that are stable inherently, there exist loss sequences that make RM$^+$ and its variants unstable, leading to cycling between very different strategies. The key reason for such instability is that the decisions of these algorithms are chosen by normalizing an *aggregate payoff vector*; thus, in a region close to the origin, two consecutive aggregate payoffs may point in very different directions, despite being close, resulting in unstable iterations. Surprisingly, note that this can only happen when the aggregate payoff vectors, which essentially measure the algorithm's regret against each action, are small, so instability can only happen when one's regret is small and thus is seemingly not an issue. However, in a game setting, such instability might cause other players to suffer large regret because they have to learn in an unpredictable environment. Indeed, we identify a $3 \times 3$ matrix game where this is the case and both RM$^+$ and predictive RM$^+$ converge slowly at a rate of $O(1/\sqrt{T})$ (Fig. 1). We emphasize that very little is known about the properties of (predictive) RM$^+$ and we are the first to show concrete examples of stability issues in matrix games and in the adversarial setting.

2. Motivated by our counterexamples, we propose two methods to stabilize RM$^+$: *restarting*, which reinitializes the algorithms when the aggregate payoffs are all below a threshold, and *chopping off* the origin from the nonnegative orthant to smooth the algorithms. When applying these techniques to online learning with RM$^+$, we show improved regret and fast convergence similar to predictive OMD: we obtain $O(T^{1/4})$ individual regrets for *Stable Predictive RM$^+$* (which uses restarting) and $O(1)$ social regret for *Smooth Predictive RM$^+$* (which chops off the origin). We also consider *conceptual prox* and *extragradient* versions of RM$^+$ for normal-form games. We show that our stabilizing ideas also provide the required stability in these settings and thus give strong theoretical guarantees: Conceptual RM$^+$ achieves $O(1)$ individual regrets (Theorem 5.3) while Extragradient RM$^+$ achieves $O(1)$ social regret (Theorem 5.6). See Table 1 for a summary of our results for normal-form games. We further extend Conceptual RM$^+$ to extensive-form games (EFG), yielding $O(1)$ regret in $T$ iterations with $O(T \log(T))$ gradient computation. The key step here is to show the Lipschitzness of the CFR decomposition (Lemma J.1).

3. We apply our algorithms to solve matrix games and EFGs. For the $3 \times 3$ matrix game instability counterexample, our algorithms indeed perform significantly better than (predictive)

RM$^+$. For random matrix games, we find that Stable and Smooth Predictive RM$^+$ have very strong empirical performance, on par with (unstabilized) Predictive RM$^+$, and greatly outperforming RM$^+$ in all our experiments; Extragradient RM$^+$ appears to be more sensitive to the choice of step sizes and sometimes performs only as well as RM$^+$. Our experiments on 4 different EFGs show that our implementation of clairvoyant CFR outperforms predictive CFR in some, but not all, instances.

## 2   Preliminaries

**Notations.** For $d \in \mathbb{N}$, we write $\mathbf{1}_d \in \mathbb{R}^d$ the vector with 1 on every component. The simplex of dimension $d - 1$ is $\Delta^d = \{\boldsymbol{x} \in \mathbb{R}^d_+ \mid \langle \boldsymbol{x}, \mathbf{1}_d \rangle = 1\}$. The vector $\mathbf{0}$ has 0 on every component and its dimension is implicit. For $x \in \mathbb{R}$, we write $[x]^+$ for the positive part of $x : [x]^+ = \max\{0, x\}$, and we overload this notation to vectors component-wise. For two vectors $\boldsymbol{a}$ and $\boldsymbol{b}$, $\boldsymbol{a} \geq \boldsymbol{b}$ means $\boldsymbol{a}$ is at least $\boldsymbol{b}$ component-wise. We write $\| \cdot \|_*$ for the dual norm of a norm $\| \cdot \|$.

**Online Linear Minimization.** In online linear minimization, at every decision period $t \geq 1$, an algorithm chooses a decision $\boldsymbol{x}^t$ from a convex decision set $\mathcal{X}$. A loss vector $\boldsymbol{\ell}^t$ is chosen arbitrarily and an instantaneous loss of $\langle \boldsymbol{\ell}^t, \boldsymbol{x}^t \rangle$ is incurred. The regret of an algorithm generating the sequence of decisions $\boldsymbol{x}^1, ..., \boldsymbol{x}^T$ is defined as the difference between the cumulative loss generated and that of any fixed strategy $\hat{\boldsymbol{x}} \in \mathcal{X}$: $\text{Reg}^T(\hat{\boldsymbol{x}}) = \sum_{t=1}^T \langle \boldsymbol{\ell}^t, \boldsymbol{x}^t - \hat{\boldsymbol{x}} \rangle$. A *regret minimizer* guarantees that $\text{Reg}^T(\hat{\boldsymbol{x}}) = o(T)$ for any $\hat{\boldsymbol{x}} \in \mathcal{X}$.

**Online Mirror Descent.** A famous regret minimizer is Online Mirror Descent (OMD) [30], which generates the decisions $\boldsymbol{x}^1, ..., \boldsymbol{x}^T$ as follows (with a learning rate $\eta > 0$):

$$\boldsymbol{x}^{t+1} = \Pi_{\boldsymbol{x}^t, \mathcal{X}}\left(\eta \boldsymbol{\ell}^t\right) \tag{OMD}$$

where for any $\boldsymbol{x} \in \mathcal{X}$, and any loss $\boldsymbol{\ell}$, the *proximal operator* $\boldsymbol{\ell} \mapsto \Pi_{\boldsymbol{x}, \mathcal{X}}(\boldsymbol{\ell})$ is defined as $\Pi_{\boldsymbol{x}, \mathcal{X}}(\boldsymbol{\ell}) = \arg\min_{\hat{\boldsymbol{x}} \in \mathcal{X}} \langle \boldsymbol{\ell}, \hat{\boldsymbol{x}} \rangle + D(\hat{\boldsymbol{x}}, \boldsymbol{x})$ where $D$ is the *Bregman divergence* associated with $\varphi : \mathcal{X} \to \mathbb{R}$, a 1-strongly convex regularizer (with respect to some norm $\| \cdot \|$): $D(\hat{\boldsymbol{x}}, \boldsymbol{x}) = \varphi(\hat{\boldsymbol{x}}) - \varphi(\boldsymbol{x}) - \langle \nabla\varphi(\boldsymbol{x}), \hat{\boldsymbol{x}} - \boldsymbol{x} \rangle, \forall \hat{\boldsymbol{x}}, \boldsymbol{x} \in \mathcal{X}$. OMD guarantees that the worst-case regret against any $\hat{\boldsymbol{x}}$ grows as $O(\sqrt{T})$ (omitting other dependence for simplicity; the same below). Other popular regret minimizers include Follow-The-Regularized-Leader (FTRL), and adaptive variants of OMD and FTRL; we refer the reader to [20] for an extensive survey on regret minimizers.

**Regret Matching and Regret Matching$^+$.** Regret Matching (RM) and Regret Matching$^+$ (RM$^+$) are two regret minimizers that achieve $O(\sqrt{T})$ worst-case regret when $\mathcal{X} = \Delta^d$. RM [19] maintains a sequence of *aggregate payoffs* $(\boldsymbol{R}^t)_{t \geq 1}$: $\boldsymbol{R}^1 = R_0 \mathbf{1}_d$, and for $t \geq 1$,

$$\boldsymbol{x}^t = [\boldsymbol{R}^t]^+ / \|[\boldsymbol{R}^t]^+\|_1, \quad \boldsymbol{R}^{t+1} = \boldsymbol{R}^t + \langle \boldsymbol{x}^t, \boldsymbol{\ell}^t \rangle \mathbf{1}_d - \boldsymbol{\ell}^t,$$

where $R_0 \geq 0$ specifies an initial point and $\mathbf{0}/0$ is defined as the uniform distribution for convenience. The original RM sets $R_0 = 0$, making the algorithm completely *parameter-free*, a very appealing property in practice. RM$^+$ is a simple variation of RM, where the aggregate payoffs are thresholded at every iteration [35]. In particular, RM$^+$ only keeps track of the non-negative components of the aggregate payoffs to compute a decision: $\boldsymbol{R}^1 = R_0 \mathbf{1}_d$, and for $t \geq 1$,

$$\boldsymbol{x}^t = \boldsymbol{R}^t / \|\boldsymbol{R}^t\|_1, \quad \boldsymbol{R}^{t+1} = [\boldsymbol{R}^t + \langle \boldsymbol{x}^t, \boldsymbol{\ell}^t \rangle \mathbf{1}_d - \boldsymbol{\ell}^t]^+.$$

We highlight that very little is known about the theoretical properties of RM$^+$, despite its strong empirical performances: [36] show that RM$^+$ is a regret minimizer (and enjoys the stronger $K$-tracking regret property), and [4] show that it can safely be combined with alternation ([18] prove strict improvement when using alternation). Farina et al. [12] show an interesting connection between RM$^+$ and Online Mirror Descent: the update $\boldsymbol{R}^{t+1} = [\boldsymbol{R}^t + \langle \boldsymbol{x}^t, \boldsymbol{\ell}^t \rangle \mathbf{1}_d - \boldsymbol{\ell}^t]^+$ of RM$^+$ can be rewritten as

$$\boldsymbol{R}^{t+1} = \Pi_{\boldsymbol{R}^t, \mathcal{X}}\left(\eta \boldsymbol{f}(\boldsymbol{x}^t, \boldsymbol{\ell}^t)\right)$$

for $\mathcal{X} = \mathbb{R}^d_+, \varphi = \frac{1}{2}\| \cdot \|_2^2, \eta = 1$, and $\boldsymbol{f}(\boldsymbol{x}^t, \boldsymbol{\ell}^t)$ defined as $\boldsymbol{f}(\boldsymbol{x}^t, \boldsymbol{\ell}^t) = \boldsymbol{\ell}^t - \langle \boldsymbol{x}^t, \boldsymbol{\ell}^t \rangle \mathbf{1}_d$. Therefore, RM$^+$ generating a sequence of decisions $\boldsymbol{x}^1, ..., \boldsymbol{x}^T$ facing a sequence of losses $(\boldsymbol{\ell}^t)_{t \geq 1}$, is closely connected to OMD instantiated with the non-negative orthant as the decision set and facing a sequence of losses $(\boldsymbol{f}(\boldsymbol{x}^t, \boldsymbol{\ell}^t))_{t \geq 1}$. We have the following relation for the regret in $\boldsymbol{x}^1, ..., \boldsymbol{x}^T$ and the regret in $\boldsymbol{R}^1, ..., \boldsymbol{R}^T$ (the proof follows [12] and is deferred to the appendix).

**Lemma 2.1.** *Let $\boldsymbol{x}^1, ..., \boldsymbol{x}^T \in \Delta^d$ be generated as $\boldsymbol{x}^t = \boldsymbol{R}^t/\|\boldsymbol{R}^t\|_1$ for some sequence $\boldsymbol{R}^1, ..., \boldsymbol{R}^T \in \mathbb{R}_+^d$. The regret $\mathrm{Reg}^T(\hat{\boldsymbol{x}})$ of $\boldsymbol{x}^1, ..., \boldsymbol{x}^T$ facing a sequence of losses $\boldsymbol{\ell}^1, ..., \boldsymbol{\ell}^T$ is equal to $\mathrm{Reg}^T(\hat{\boldsymbol{R}})$, the regret of $\boldsymbol{R}^1, ..., \boldsymbol{R}^T$ facing the sequence of losses $\boldsymbol{f}\left(\boldsymbol{x}^1, \boldsymbol{\ell}^1\right), ..., \boldsymbol{f}\left(\boldsymbol{x}^T, \boldsymbol{\ell}^T\right)$, compared against $\hat{\boldsymbol{R}} = \hat{\boldsymbol{x}}$: $\mathrm{Reg}^T\left(\hat{\boldsymbol{R}}\right) = \sum_{t=1}^T \left\langle \boldsymbol{f}\left(\boldsymbol{x}^t, \boldsymbol{\ell}^t\right), \boldsymbol{R}^t - \hat{\boldsymbol{R}}\right\rangle.$*

Since OMD is a regret minimizer guaranteeing $\mathrm{Reg}^T(\hat{\boldsymbol{R}}) = O(\sqrt{T})$, Lemma 2.1 directly shows that RM$^+$ is also a regret minimizer: $\mathrm{Reg}^T(\hat{\boldsymbol{x}}) = O(\sqrt{T})$.

**Multiplayer Normal-Form Games.** In a multiplayer normal-form game, there are $n \in \mathbb{N}$ players. Each player $i$ has $d_i$ strategies and their decision space $\Delta^{d_i}$ is the probability simplex over the $d_i$ strategies. We denote $\Delta = \times_{i=1}^n \Delta^{d_i}$ as the joint decision space of all players and $d = d_1 + \cdots + d_n$ as the joint decision space of all players. The utility function for player $i$ is a concave function $u_i : \Delta \to [-1, 1]$ that maps every joint strategy profile $\boldsymbol{x} = (\boldsymbol{x}_1, ..., \boldsymbol{x}_n) \in \Delta$ to a payoff. We assume bounded gradients and $L_u$-smoothness for the utilities of the players: there exists $B_u > 0, L_u > 0$ such that for any $\boldsymbol{x}, \boldsymbol{x}' \in \Delta$ and any player $i$,

$$\|\nabla_{\boldsymbol{x}_i} u_i(\boldsymbol{x})\|_2 \le B_u, \|\nabla_{\boldsymbol{x}_i} u_i(\boldsymbol{x}) - \nabla_{\boldsymbol{x}_i} u_i(\boldsymbol{x}')\|_2 \le L_u \|\boldsymbol{x} - \boldsymbol{x}'\|_2. \tag{1}$$

The function mapping joint strategies to negative payoff gradients for all players is a vector-valued function $G : \Delta \to \mathbb{R}^d$ such that $G(\boldsymbol{x}) = (-\nabla_{\boldsymbol{x}_1} u_1(\boldsymbol{x}), \ldots, -\nabla_{\boldsymbol{x}_n} u_n(\boldsymbol{x}))$. It is well known that running a regret minimizer for $(\boldsymbol{x}_1^t, ..., \boldsymbol{x}_n^t) \in \Delta = \times_{i=1}^n \Delta^{d_i}$ facing the loss $G(\boldsymbol{x}^t) = (\boldsymbol{\ell}_1^t, \ldots, \boldsymbol{\ell}_n^t)$ leads to strong game-theoretic guarantees (e.g., the average iterate being an approximate coarse correlated equilibrium). However, in light of Lemma 2.1, we will instead perform regret minimization on $(\boldsymbol{R}_1^t, ..., \boldsymbol{R}_n^t) \in \mathcal{X} = \times_{i=1}^n \mathbb{R}_+^{d_i}$ with the losses $(\boldsymbol{f}(\boldsymbol{x}_1^t, \boldsymbol{\ell}_1^t), \ldots, \boldsymbol{f}(\boldsymbol{x}_n^t, \boldsymbol{\ell}_n^t))$. For conciseness, we thus define the operator $F : \mathcal{X} \to \mathbb{R}^d$ as, for $\boldsymbol{z} = (\boldsymbol{R}_1, ..., \boldsymbol{R}_n)$, $F(\boldsymbol{z}) = (\boldsymbol{f}(\boldsymbol{x}_1, \boldsymbol{\ell}_1), ..., \boldsymbol{f}(\boldsymbol{x}_n, \boldsymbol{\ell}_n))$ where $\boldsymbol{x}_i = \boldsymbol{R}_i/\|\boldsymbol{R}_i\|_1, \forall\, i = 1, ..., n, (\boldsymbol{\ell}_i)_{i \in [n]} = G(\boldsymbol{x})$.

**Predictive OMD and Its RVU Bounds.** The predictive version of OMD proceeds as follows:

$$\boldsymbol{x}^t = \Pi_{\tilde{\boldsymbol{x}}^t, \mathcal{X}}\left(\eta \boldsymbol{m}^t\right) \quad \tilde{\boldsymbol{x}}^{t+1} = \Pi_{\tilde{\boldsymbol{x}}^t, \mathcal{X}}\left(\eta \boldsymbol{\ell}^t\right)$$

When setting $\boldsymbol{m}^t = \boldsymbol{\ell}^{t-1}$, predictive OMD satisfies $\mathrm{Reg}^T(\hat{\boldsymbol{x}}) \le \frac{D(\hat{\boldsymbol{x}}, \tilde{\boldsymbol{x}}^1)}{\eta} + \eta \sum_{t=1}^T \|\boldsymbol{\ell}^t - \boldsymbol{\ell}^{t-1}\|_*^2 - \frac{1}{8\eta} \sum_{t=1}^{T-1} \|\boldsymbol{x}^{t+1} - \boldsymbol{x}^t\|^2$. This regret bound satisfies the *RVU* (regret bounded by variation in utilities) condition, introduced in [34]. The authors show that this type of bound guarantees that the social regret (i.e., sum of the regrets of all players) is $O(1)$ when all players apply this special instance of predictive OMD. Syrgkanis et al. [34] further prove that each player has improved $O(T^{1/4})$ individual regret by the *stability* of predictive OMD. Specifically, they show that predictive OMD guarantees $\|\boldsymbol{x}^{t+1} - \boldsymbol{x}^t\| = O(\eta)$ against any adversarial loss sequence, i.e., the algorithm is stable in the sense that the change in the iterates can be controlled by choosing $\eta$ appropriately.

**Predictive RM$^+$** Similar to OMD, we can generalize RM$^+$ to Predictive Regret Matching$^+$[12]: define $\boldsymbol{R}^1 = \boldsymbol{m}^1 = R_0 \mathbf{1}_d$ (with $R_0 = 0$ by default), and for $t \ge 1$,

$$\boldsymbol{x}^t = \hat{\boldsymbol{R}}^t/\|\hat{\boldsymbol{R}}^t\|_1, \text{ for } \hat{\boldsymbol{R}}^t = [\boldsymbol{R}^t + \boldsymbol{m}^t]^+,$$
$$\boldsymbol{R}^{t+1} = [\boldsymbol{R}^t - \boldsymbol{f}(\boldsymbol{x}^t, \boldsymbol{\ell}^t)]^+, \text{ for } \boldsymbol{f}(\boldsymbol{x}^t, \boldsymbol{\ell}^t) = \boldsymbol{\ell}^t - \left\langle \boldsymbol{x}^t, \boldsymbol{\ell}^t \right\rangle \mathbf{1}_d.$$

We call the algorithm predictive RM$^+$ (PRM$^+$) when $\boldsymbol{m}^t = -\boldsymbol{f}(\boldsymbol{x}^{t-1}, \boldsymbol{\ell}^{t-1})$, and it recovers RM$^+$ when $\boldsymbol{m}^t = \mathbf{0}$. A regret bound with a similar RVU condition is attainable for predictive RM$^+$ by its connection to predictive OMD [12], but only in the non-negative orthant space instead of the actual strategy space. To make a connection between them, stability is required as we show later. A natural question is then whether (predictive) RM$^+$ is also always stable. We show that the answer is no by giving an adversarial example in the next section.

# 3 Instability of (Predictive) Regret Matching$^+$

We start by showing that there exist adversarial loss sequences that lead to instability for both RM$^+$ and predictive RM$^+$. Our construction starts with an unbounded loss sequence $\boldsymbol{\ell}^t$ so that $\boldsymbol{x}^t$ alternates between $(1/2, 1/2)$ and $(0, 1)$: we set $\boldsymbol{\ell}^t = (\ell^t, 0)$, where $\ell^1 = 2$, and for $t \ge 2$, $\ell^t = -2^{(t-2)/2}$ if $t$ is even and $\ell^t = 2^{(t-1)/2}$ if $t$ is odd. Our proof is completed by normalizing the losses to $[-1, 1]$ given a fixed time horizon (see Appendix B for details).

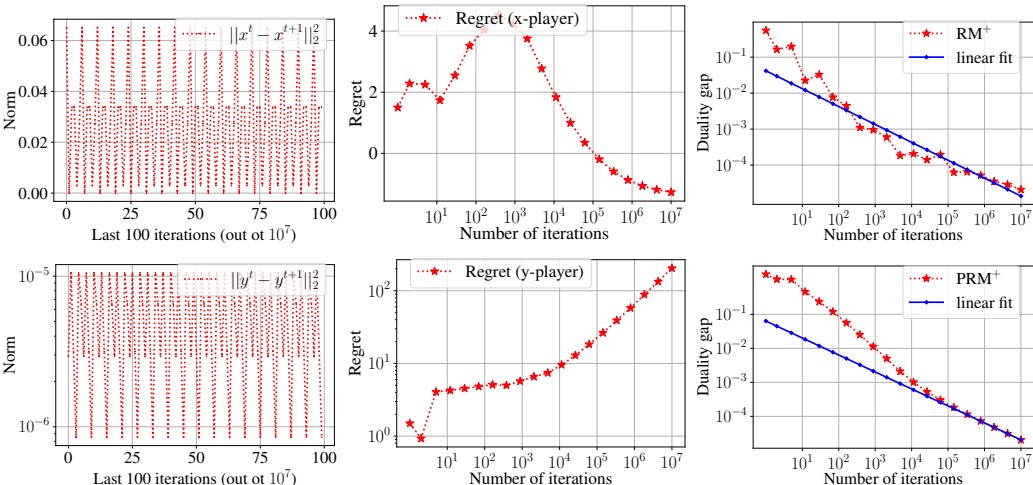

Figure 1: Left plots show the iterate-to-iterate variation in the last 100 iterates of predictive RM$^+$. Center plots show the regret for the $x$ and $y$ players under predictive RM$^+$. Right plots show empirical convergence speed of RM$^+$ (top row) and Predictive RM$^+$ (bottom row).

**Theorem 3.1.** *There exist finite sequences of losses in $\mathbb{R}^2$ for RM$^+$ and its predictive version such that $\boldsymbol{x}^t = (\frac{1}{2}, \frac{1}{2})$ when $t$ is odd and $\boldsymbol{x}^t = (0, 1)$ when $t$ is even.*

This is in stark contrast to OMD which always ensures $\|\boldsymbol{x}^{t+1} - \boldsymbol{x}^t\| = O(\eta)$ and is thus inherently stable. However, a somewhat surprising property about (predictive) RM$^+$ is that *instability actually implies low regret*. To see this, we first present the following Lipschitz property of the normalization function $\boldsymbol{g} : \boldsymbol{x} \mapsto \boldsymbol{x}/\|\boldsymbol{x}\|_1$ for $\boldsymbol{x} \in \mathbb{R}^d_+$.

**Proposition 1.** Let $\boldsymbol{x}, \boldsymbol{y} \in \mathbb{R}^d_+$, with $\mathbf{1}^\top \boldsymbol{x} \geq 1$. Then, $\|\boldsymbol{g}(\boldsymbol{y}) - \boldsymbol{g}(\boldsymbol{x})\|_2 \leq \sqrt{d} \cdot \|\boldsymbol{y} - \boldsymbol{x}\|_2$.

This proposition shows that the normalization step has a reasonable Lipschitz constant ($\sqrt{d}$) as long as its input is not too close to the origin, which further implies the following corollary.

**Corollary 3.2.** RM$^+$ with $\|\boldsymbol{R}^t\|_1 \geq R_0$ satisfies $\left\|\boldsymbol{x}^{t+1} - \boldsymbol{x}^t\right\|_2 \leq \frac{\sqrt{d}}{R_0} \cdot \left\|\boldsymbol{R}^{t+1} - \boldsymbol{R}^t\right\|_2 \leq \frac{2dB_u}{R_0}$.

Put differently, the corollary states that instability can happen only when the cumulative regret vector $\boldsymbol{R}^t$ is small. For example, if $\|\boldsymbol{x}^{t+1} - \boldsymbol{x}^t\| = \Omega(1)$, then we must have $\|\boldsymbol{R}^t\|_1 = O(dB_u)$ and thus the regret at that point is at most $O(dB_u)$. A similar argument holds for predictive RM$^+$ as well. Therefore, instability is in fact not an issue for these algorithms' own regret.

However, when using these algorithms to play a game, what could happen is that such instability leads to other players learning in an unpredictable environment with large regret. We show this phenomenon via an example of a $3 \times 3$ matrix game $\max_{\boldsymbol{x} \in \Delta(3)} \min_{\boldsymbol{y} \in \Delta(3)} \langle \boldsymbol{x}, \boldsymbol{A}\boldsymbol{y} \rangle$, where $\boldsymbol{A} = ((3, 0, -3), (0, 3, -4), (0, 0, 1))$. The first column of Fig. 1 shows the squared $\ell_2$ norm of the consecutive difference of the last 100 iterates of Predictive RM$^+$ for the $x$ player (top) and the $y$ player (bottom). The iterates of the $x$ player are rapidly changing in a periodic fashion while the iterates of the $y$ player are stable with changes on the order of $10^{-5}$. In the center plots where we show the individual regret for each player, we indeed observe that the cumulative regret of the $x$ player is near zero as implied by instability, but it causes large regret (close to $T^{0.5}$ empirically) for the $y$ player. (We show the same plots for RM$^+$ in Fig. 4 in Appendix B; there, the iterates of both players are stable, but since RM$^+$ lacks predictivity, it still leads to larger regret for one player.)

The right column of Fig. 1 shows the duality gap achieved by the linear average $(\bar{x}_t, \bar{y}_t) = \left( \frac{2}{T(T+1)} \sum_{t=1}^T t\boldsymbol{x}^t, \frac{2}{T(T+1)} \sum_{t=1}^T t\boldsymbol{y}^t \right)$, when the iterates are generated by RM$^+$ with alternation (top) and predictive RM$^+$ (bottom). For both algorithms the convergence rate slows down around $10^4$ iterations. A linear regression estimate on the rate for the last $10^6$ iterates shows rates of $-0.497$ and $-0.496$ for RM$^+$ and predictive RM$^+$ respectively. To the best of our knowledge, this is the

first known case of empirical convergence rates on the order of $T^{-0.5}$ for either RM$^+$ or predictive RM$^+$; the worst prior instance for RM$^+$ was $T^{-0.74}$ in Farina et al. [10]; no hard instance was known for predictive RM$^+$.

## 4 Stabilizing RM$^+$ and Predictive RM$^+$.

Based on the discussions in the previous section, we aim to make *every* player stable despite the fact that being an unstable player may actually be good for that particular player. By Corollary 3.2, it suffices to make sure that $\|\boldsymbol{R}^t\|_1$ is never too small. We provide two approaches to ensure this property and thereby stabilize (predictive) RM$^+$.

**Stable Predictive RM$^+$.** One way to maintain the required distance to the origin is via *restarting*: We initialize the algorithm with the cumulative regret vector equal to some non-zero amount, instead of the usual initialization at zero. Then, when the cumulative regret vector gets below the initialization point, we *restart* the algorithm from the initialization point. Applying this idea to predictive RM$^+$ yields Algorithm 1. Player $i$ starts with $\boldsymbol{R}_i^1 = R_0 \mathbf{1}_{d_i}$, runs predictive RM$^+$, and restarts whenever $\boldsymbol{R}_i^t \leq R_0 \mathbf{1}_{d_i}$. In the algorithm we write $(\boldsymbol{R}_1^t, ..., \boldsymbol{R}_n^t)$ compactly as $\boldsymbol{w}^t$ (similarly for $\boldsymbol{z}^t$). Note, though, that the updates are decentralized for each player, as in vanilla predictive RM$^+$.

Given this modification, Stable PRM$^+$ achieves improved individual regret in multiplayer games, as stated in Theorem 4.1. We defer the proof to the appendix. One key step in the analysis is to note that by definition, the regret against any action is negative when the restarting event happens, so it is sufficient to consider the regret starting from the last restart. Thanks to the stability enforced by the restarts, the regret from the last restart is also well controlled and the results follow by tuning $\eta$ and $R_0$ optimally. In fact, since the algorithm is scale-invariant up to the relative scale of the two parameters, it is without loss of generality to always set $R_0 = 1$.

---

**Algorithm 1** Stable Predictive RM$^+$

1: **Input**: $R_0 > 0$, step size $\eta > 0$
2: **Initialization**: $\boldsymbol{w}^0 = R_0 \mathbf{1}_d$
3: **for** $t = 1, \ldots, T$ **do**
4:      $\boldsymbol{z}^t = \Pi_{\boldsymbol{w}^{t-1}, \mathcal{X}} \left( \eta \boldsymbol{m}^t \right)$
5:      $\boldsymbol{w}^t = \Pi_{\boldsymbol{w}^{t-1}, \mathcal{X}} \left( \eta F(\boldsymbol{z}^t) \right)$
6:      $(\boldsymbol{x}_1^t, \ldots, \boldsymbol{x}_n^t) = (\boldsymbol{g}(\boldsymbol{z}_1^t), \ldots, \boldsymbol{g}(\boldsymbol{z}_n^t))$
7:      **for** $i = 1, ..., n$ **do**
8:          **if** $\boldsymbol{w}_i^t \leq R_0 \mathbf{1}_{d_i}$ **then**
9:              $\boldsymbol{w}_i^t = R_0 \mathbf{1}_{d_i}$

**Algorithm 2** Smooth Predictive RM$^+$

1: **Input**: Step size $\eta > 0$
2: **Initialization**: $\boldsymbol{w}^0 \in \mathcal{X}_{\geq}$
3: **for** $t = 1, \ldots, T$ **do**
4:      $\boldsymbol{z}^t = \Pi_{\boldsymbol{w}^{t-1}, \mathcal{X}_{\geq}} \left( \eta \boldsymbol{m}^t \right)$
5:      $\boldsymbol{w}^t = \Pi_{\boldsymbol{w}^{t-1}, \mathcal{X}_{\geq}} \left( \eta F(\boldsymbol{z}^t) \right)$
6:      $(\boldsymbol{x}_1^t, \ldots, \boldsymbol{x}_n^t) = (\boldsymbol{g}(\boldsymbol{z}_1^t), \ldots, \boldsymbol{g}(\boldsymbol{z}_n^t))$

---

**Theorem 4.1.** *Let* $\eta = \left( d^2 T \right)^{-1/4}$ *and* $R_0 = 1$. *Let* $(\boldsymbol{f}_i^t)_{i \in [n]} = F(\boldsymbol{z}^t)$ *for* $t \geq 1$. *For each player* $i$, *set the sequence of predictions* $\boldsymbol{m}_i^t = \boldsymbol{0}$ *when* $t = 0$ *or restart happens at* $t - 1$; *otherwise,* $\boldsymbol{m}_i^t = \boldsymbol{f}_i^{t-1}, \forall\ t \geq 1$. *Then Algorithm 1 guarantees that the individual regret* $\text{Reg}_i^T(\hat{\boldsymbol{x}}_i) = \sum_{t=1}^T \langle \nabla_{\boldsymbol{x}_i} u_i(\boldsymbol{x}^t), \hat{\boldsymbol{x}}_i - \boldsymbol{x}_i^t \rangle$ *of each player* $i$ *is bounded by* $O\left( d^{3/2} T^{1/4} \right)$ *in multiplayer normal-form games.*

Although the restarting idea successfully stabilizes the RM$^+$ algorithm, the discontinuity created by asynchronous restarts causes technical difficulty for bounding the social regret by $O(1)$. Next we introduce an alternative stabilization idea to fix this issue.

**Smooth Predictive RM$^+$.** Our second stabilization idea is to restrict the decision space to a subset where we "chop off" the area that is too close to the origin, that is, project the vector $\boldsymbol{R}_i^t$ onto the set $\Delta_{\geq}^{d_i} = \{\boldsymbol{R} \in \mathbb{R}_+^{d_i} \mid \|\boldsymbol{R}\|_1 \geq 1\}$. We denote the joint chopped-off decision space as $\mathcal{X}_{\geq} = \times_{i=1}^n \Delta_{\geq}^{d_i}$. We call the resulting algorithm smooth predictive RM$^+$ (Algorithm 2). Besides a similar result to Theorem 4.1 on the individual regret (omitted for simplicity), Algorithm 2 also guarantees that the social regret is bounded by a game-dependent constant, as shown in Theorem 4.2.

**Theorem 4.2.** *Let* $\eta = \left( 2\sqrt{2}(n-1) \max_i \{d_i^{3/2}\} \right)^{-1}$. *Using the sequence of predictions* $\boldsymbol{m}^0 = \boldsymbol{0}, \boldsymbol{m}^t = F(\boldsymbol{z}^{t-1}), \forall\ t \geq 1$, *Algorithm 2 guarantees that the so-*

| **Algorithm 3** Conceptual RM$^+$ | **Algorithm 4** Conceptual RM$^+$ with approximate fixed-point |
|---|---|
| 1: **Input**: Step size $\eta > 0$ with $\eta < 1/L_F$ | |
| 2: **Initialization**: $\boldsymbol{z}^0 \in \mathcal{X}_{\geq}$ | 1: **Input**: Step size $\eta > 0$ with $\eta < 1/L_F$ |
| 3: **for** $t = 1, \ldots, T$ **do** | 2: **Initialization**: $\boldsymbol{z}^0 \in \mathcal{X}_{\geq}$ |
| 4: $\quad \boldsymbol{z}^t = \Pi_{\boldsymbol{z}^{t-1}, \mathcal{X}_{\geq}} \left( \eta F(\boldsymbol{z}^t) \right)$ | 3: **for** $t = 1, \ldots, T$ **do** |
| 5: $\quad (\boldsymbol{x}_1^t, \ldots, \boldsymbol{x}_n^t) = (\boldsymbol{g}(\boldsymbol{z}_1^t), \ldots, \boldsymbol{g}(\boldsymbol{z}_n^t))$ | 4: $\quad \boldsymbol{w}^0 = \boldsymbol{z}^{t-1}$ |
| | 5: $\quad$ **for** $j = 0, \ldots, k-1$ **do** |
| | 6: $\qquad \boldsymbol{w}^{j+1} = \Pi_{\boldsymbol{z}^{t-1}, \mathcal{X}_{\geq}} \left( \eta F(\boldsymbol{w}^j) \right)$ |
| | 7: $\quad \boldsymbol{z}^t = \Pi_{\boldsymbol{z}^{t-1}, \mathcal{X}_{\geq}} \left( \eta F(\boldsymbol{w}^k) \right)$ |
| | 8: $\quad (\boldsymbol{x}_1^t, \ldots, \boldsymbol{x}_n^t) = \left( \boldsymbol{g}(\boldsymbol{w}_1^k), \ldots, \boldsymbol{g}(\boldsymbol{w}_n^k) \right)$ |

cial regret $\sum_{i=1}^n \text{Reg}_i^T(\hat{\boldsymbol{x}}_i) = \sum_{i=1}^n \sum_{t=1}^T \langle \nabla_{\boldsymbol{x}_i} u_i(\boldsymbol{x}^t), \hat{\boldsymbol{x}}_i - \boldsymbol{x}_i^t \rangle$ is upper bounded by $O\left( n^2 \max_{i=1,\ldots,n} \{d_i^{3/2}\} \max_{i=1,\ldots,n} \{\|\boldsymbol{w}_i^0 - \hat{\boldsymbol{x}}_i\|_2^2\} \right)$ in multiplayer normal-form games.

Algorithm 2 dominates Algorithm 1 in terms of our theoretical results so far, but it has one drawback: it requires occasional projection onto $\mathcal{X}_{\geq}$. In Appendix K we show that this can be done with a sorting trick in $O(d \log d)$ time, whereas the restarting procedure is implementable in linear time.

## 5 Conceptual Regret Matching$^+$

In this section, we depart from the predictive OMD framework and develop new smooth variants of RM$^+$ from a different angle. Instead of using predictive OMD to compute the iterates $(\boldsymbol{R}_i^t)_{t \geq 1}$, we consider the following regret minimizer that we call *cheating OMD*, defined for some arbitrary closed decision set $\mathcal{Z}$ and an arbitrary sequence of losses $(\boldsymbol{\ell}^t)_{t \geq 1}$: $\boldsymbol{z}^t = \Pi_{\boldsymbol{z}^{t-1}, \mathcal{Z}} \left( \eta \boldsymbol{\ell}^t \right)$ for $t \geq 1$, and $\boldsymbol{z}^0 \in \mathcal{X}_{\geq}$. Cheating OMD is inspired by the Conceptual Prox method for solving variational inequalities associated with monotone operators [5, 23, 29]. We call it *cheating* OMD because at iteration $t$, the decision $\boldsymbol{z}^t$ is chosen as a function of the current loss $\boldsymbol{\ell}^t$, which is revealed *after* the decision $\boldsymbol{z}^t$ has been chosen. It is well-known that cheating OMD yields a sequence of decisions with constant regret; we show it for our setting in the following lemma.

**Lemma 5.1.** *The Cheating OMD iterates $\{\boldsymbol{z}^t\}_t$ satisfy $\sum_{t=1}^T \langle \boldsymbol{\ell}^t, \boldsymbol{z}^t - \hat{\boldsymbol{z}} \rangle \leq \frac{1}{2\eta} \|\boldsymbol{z}^0 - \hat{\boldsymbol{z}}\|_2^2, \forall \hat{\boldsymbol{z}} \in \mathcal{Z}$.*

To instantiate RM$^+$ with Cheating OMD as a regret minimizer for the sequence $(\boldsymbol{R}_i^t)_{t \geq 1}$ of each player $i$, we need to show the existence of a vector $\boldsymbol{z}^t \in \mathcal{X}_{\geq}$ such that

$$\boldsymbol{z}^t = \Pi_{\boldsymbol{z}^{t-1}, \mathcal{X}_{\geq}} \left( \eta F(\boldsymbol{z}^t) \right). \tag{2}$$

Equation (2) can be interpreted as a fixed-point equation for the map $\boldsymbol{z} \mapsto \Pi_{\boldsymbol{z}^{t-1}, \mathcal{X}_{\geq}} (\eta F(\boldsymbol{z}))$. For any $\boldsymbol{z}' \in \mathcal{X}_{\geq}$, the map $\boldsymbol{z} \mapsto \Pi_{\boldsymbol{z}', \mathcal{X}_{\geq}} (\eta F(\boldsymbol{z}))$ is $\eta L$-Lipschitz continuous *as long as* $F$ is $L$-Lipschitz continuous. Therefore, it is a contraction when $\eta < 1/L$, and then the fixed-point equation $\boldsymbol{z} = \Pi_{\boldsymbol{z}', \mathcal{X}_{\geq}} (\eta F(\boldsymbol{z}))$ has a unique solution. Recall that for $\boldsymbol{z} = (\boldsymbol{R}_1, ..., \boldsymbol{R}_n) \in \mathcal{X}_{\geq}$, the operator $F$ is defined as $F(\boldsymbol{z}) = (\boldsymbol{f}(\boldsymbol{x}_1, \boldsymbol{\ell}_1), \ldots, \boldsymbol{f}(\boldsymbol{x}_n, \boldsymbol{\ell}_n))$ where $\boldsymbol{x}_i = \boldsymbol{g}(\boldsymbol{R}_i)$ and $\boldsymbol{\ell}_i = -\nabla_{\boldsymbol{x}_i} u_i(\boldsymbol{x})$, for all $i \in \{1, ..., n\}$. We now show the Lipschitzness of $F$ over $\mathcal{X}_{\geq}$ for normal-form games.

**Lemma 5.2.** *For a normal-form game, the operator $F$ is $L_F$-Lipschitz continuous over $\mathcal{X}_{\geq}$, with $L_F = (\max_i d_i) \sqrt{2B_u^2 + 4L_u^2}$ with $B_u, L_u$ defined in (1).*

For $L_F$ defined as in Lemma 5.2 and $\eta < 1/L_F$, the existence of the fixed-point $\boldsymbol{z}^t = \Pi_{\boldsymbol{z}^{t-1}, \mathcal{X}_{\geq}} (\eta F(\boldsymbol{z}^t))$ is guaranteed. This yields *Conceptual RM$^+$*, defined in Algorithm 3. In the following theorem, we show that Conceptual RM$^+$ ensures constant regret for each player.

**Theorem 5.3.** *Let $L_F > 0$ be defined as in Lemma 5.2. For $\eta < 1/L_F$, Algorithm 3 guarantees that the individual regret $\text{Reg}_i^T(\hat{\boldsymbol{x}}_i) = \sum_{t=1}^T \langle \nabla_{\boldsymbol{x}_i} u_i(\boldsymbol{x}^t), \hat{\boldsymbol{x}}_i - \boldsymbol{x}_i^t \rangle$ of each player $i$ is bounded by $\frac{1}{2\eta} \|\boldsymbol{z}_i^0 - \hat{\boldsymbol{x}}_i\|_2^2$ in multiplayer normal-form games.*

Note that the requirement of $\eta < 1/L_F$ in Theorem 5.3 and Algorithm 3 is only needed in order to ensure existence of a fixed-point. If the fixed-point condition holds for some larger $\eta$, then the algorithm is still well-defined and the same convergence guarantee holds.

**Remark 5.4.** *Piliouras et al. [31] propose the* clairvoyant multiplicate weights updates *(MWU) algorithm, based on the classical MWU algorithm, but where the rescaling at iteration $t$ involves the payoff of the players at iteration $t$. The connection with the conceptual prox method is made explicit by [14], where they show how to extend clairvoyant MWU for normal-form games to* clairvoyant OMD *for general convex games. Our algorithm uses the same idea but for $RM^+$.*

For $z' \in \mathcal{X}_\geq$, we can approximate the fixed-point of $z \mapsto \Pi_{z',\mathcal{X}_\geq}(\eta F(z))$ by performing $k \in \mathbb{N}$ fixed-point iterations. This results in Algorithm 4. We give the guarantees for Algorithm 4 below.

**Theorem 5.5.** *Let $L_F > 0$ be defined as in Lemma 5.2 and $\eta < 1/L_F$. Assume that in Algorithm 4, we ensure $\|w^k - \Pi_{z^{t-1},\mathcal{X}_\geq}(\eta F(w^k))\|_2 \leq \epsilon^{(t)}$, for all $t \geq 1$. Then Algorithm 4 guarantees that the individual regret $\mathrm{Reg}_i^T(\hat{x}_i) = \sum_{t=1}^T \langle \nabla_{x_i} u_i(x^t), \hat{x}_i - x_i^t \rangle$ of each player $i$ is bounded by $\frac{1}{2\eta}\|z_i^0 - \hat{x}_i\|_2^2 + 2B_u\sqrt{d_i}\sum_{t=1}^T \epsilon^{(t)}$ in multiplayer normal-form games.*

By Theorem 5.5, if we ensure error $\epsilon^{(t)} = 1/t^2$ in Algorithm 4 then the individual regret of each player is bounded by a constant. Since $w \mapsto \Pi_{z^{t-1},\mathcal{X}_\geq}(\eta F(w))$ is a contraction for $\eta < 1/L_F$, this only requires $k = O(\log(t))$ fixed-point iterations at each time $t$. If the number of iterations $T$ is known in advance, we can choose $k = O(\log(T))$, to ensure $\epsilon^{(t)} = O(1/T)$ and therefore that the individual regret of each player $i$ is bounded by the constant $\frac{1}{2\eta}\|z_i^0 - \hat{x}_i\|_2^2 + O(2B_u\sqrt{d_i})$.

Recall that the uniform distribution over a sequence of strategy profiles $\{x^t\}_{t=1}^T$ is a $(\max_i \mathrm{Reg}_i^T)/T$-approximate coarse correlated equilibrium (CCE) of a multiplayer normal-form game (see e.g. Theorem 2.4 in Piliouras et al. [31]). Therefore, Algorithm 3 guarantees $O(1/T)$ convergence to a CCE after $T$ iterations. With the setup from Theorem 5.5 and $k = O(\log(T))$, Algorithm 4 guarantees $O(\log(T)/T)$ convergence to a CCE after $T$ evaluations of the operator $F$.

**Extragradient RM$^+$.** We now consider the case of Algorithm 4 but with only one fixed-point iteration ($k = 1$). This is similar to the mirror prox algorithm [29] or the extragradient method [24]. We call this algorithm extragradient RM$^+$(ExRM$^+$, Algorithm 5). We show that one fixed-point iteration ($k = 1$) at every iteration ensures constant social regret.

---
**Algorithm 5** Extragradient RM$^+$ (ExRM$^+$)

1: **Input**: Step size $\eta > 0$ with $\eta < 1/L_F$
2: **Initialization**: $z^0 \in \mathcal{X}_\geq$
3: **for** $t = 1, \ldots, T$ **do**
4: $\quad w^t = \Pi_{z^{t-1},\mathcal{X}_\geq}(\eta F(z^{t-1}))$
5: $\quad z^t = \Pi_{z^{t-1},\mathcal{X}_\geq}(\eta F(w^t))$
6: $\quad (x_1^t, \ldots, x_n^t) = (g(w_1^t), \ldots, g(w_n^t))$
---

**Theorem 5.6.** *Define $L_F$ as in Lemma 5.2 and let $\eta = (\sqrt{2}L_F)^{-1}$. Algorithm 5 guarantees that the social regret $\sum_{i=1}^n \mathrm{Reg}_i^T(\hat{x}_i) = \sum_{i=1}^n \sum_{t=1}^T \langle \nabla_{x_i} u_i(x^t), \hat{x}_i - x_i^t \rangle$ is bounded by $\frac{1}{2\eta}\sum_{i=1}^n \|z_i^0 - \hat{x}_i\|_2^2$ in multiplayer normal-form games.*

We now apply Theorem 5.6 to the case of matrix games, where the goal is to solve

$$\min_{x \in \Delta^{d_1}} \max_{y \in \Delta^{d_2}} \langle x, Ay \rangle$$

for $A^{d_1 \times d_2}$. The operator $F$ is defined as

$$F\begin{bmatrix} R_1 \\ R_2 \end{bmatrix} = \begin{bmatrix} f(g(R_1), Ag(R_2)) \\ f(g(R_2), -A^\top g(R_1)) \end{bmatrix}$$

and $\mathcal{X}_\geq = \Delta_\geq^{d_1} \times \Delta_\geq^{d_2}$. The next lemma gives the Lipschitz constant of the operator $F$ in the case of matrix games.

**Lemma 5.7.** *For matrix games, the operator $F$ is $L_F$-Lipschitz over $\mathcal{X}_\geq$, with $L_F = \sqrt{6}\|A\|_{op}\max\{d_1, d_2\}$ with $\|A\|_{op} = \sup\{\|Av\|_2/\|v\|_2 \mid v \in \mathbb{R}^{d_2}, v \neq 0\}$.*

Combining Lemma 5.7 with Theorem 5.6, ExRM$^+$ for matrix games with $\mathcal{X}_\geq$ as a decision set and $\eta = (\sqrt{2}L_F)^{-1}$ guarantees constant social regret, so that the average of the iterates computed by ExRM$^+$ converges to a Nash Equilibrium at a rate of $O(1/T)$ [16].

**Extensive-form games** Our convergence results for Conceptual RM$^+$ apply beyond normal-form games, to EFGs. Briefly, a EFG is a game played on a tree, where each node belongs to some player, and the player chooses a probability distribution over branches. Moreover, players have *information sets*, which are groups of nodes belonging to a player such that they cannot distinguish among them, and thus they must choose the same probability distribution at all nodes in an information set. As is standard, we assume that each player never forgets information. Below, we describe the main ideas behind the extension; details are given in Appendix J.

In order to extend our results, we use the CFR regret decomposition [37, 9]. CFR defines a notion of local regret at each information set, using so-called *counterfactual values*. By minimizing the regret incurred at each information set with respect to counterfactual values, CFR guarantees that the overall regret over tree-form strategies is minimized. Importantly, counterfactual values are multilinear in the strategies of the players, and therefore they are Lipschitz functions of the strategies of the other players. Hence, using Algorithm 4 at each information set with counterfactual value and applying Theorem 5.5 begets a smooth-RM$^+$-based algorithm that computes a sequence of iterates with regret at most $\epsilon$ in at $O(1/\epsilon)$ iterations and using $O(\log(1/\epsilon)/\epsilon)$ gradient computations.

## 6 Numerical experiments

**Matrix games.** We compute the performance of ExRM$^+$, Stable and Smooth PRM$^+$ on the $3 \times 3$ matrix game instance from Section 2 (with step size $\eta = 0.1$) and on 30 random matrix games of size $(d_1, d_2) = (30, 40)$ with normally distributed coefficients of the payoff matrix and with step sizes $\eta \in \{0.1, 1, 10\}$. We initialize our algorithms at $(1/d_1)\mathbf{1}_d$, all algorithms use linear averaging, and all algorithms (except ExRM$^+$) use alternation. The results are shown in Figure 2. Our new algorithms greatly outperform RM$^+$ and PRM$^+$ in the $3 \times 3$ matrix game; linear regression finds an asymptotic convergence rate of $O(1/T^2)$. More detailed results for this instance are given in Appendix K.1. For random matrix games, our algorithms ExRM$^+$, Smooth PRM$^+$ and Stable PRM$^+$ all outperform RM$^+$ for stepsize $\eta = 0.1$. ExRM$^+$ performs on par with RM$^+$ for larger values of $\eta$, while Stable PRM$^+$ and Smooth PRM$^+$ remain very competitive, performing on par with the unstabilized version of PRM$^+$. We note that we use step sizes that are larger than the theoretical ones since the latter may be overly conservative [10, 25].

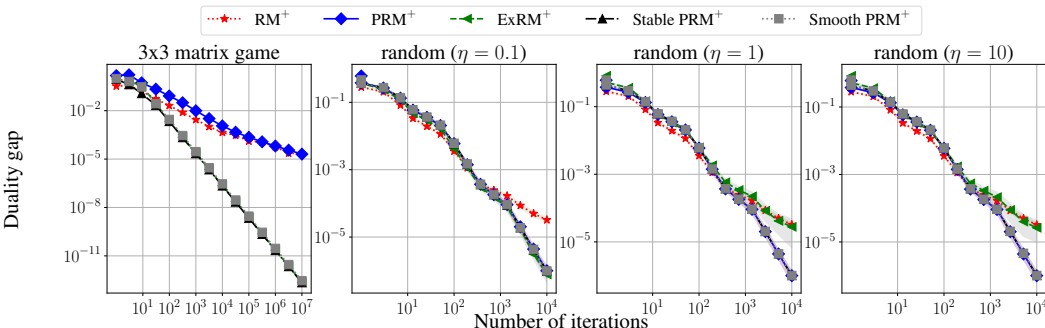

Figure 2: Empirical performances of RM$^+$, PRM$^+$, ExRM$^+$, Stable PRM$^+$ and Smooth PRM$^+$on our $3 \times 3$ matrix game (left plot) and on random instances for different step sizes.

**Extensive-form games.** We implemented and evaluated our CFR-based clairvoyant algorithm (henceforth 'Clairvoyant CFR') for extensive-form games. To our knowledge, it is the first time that clairvoyant algorithms are evaluated in extensive-form games. Overall, we were unable to observe the same strong performance observed in normal-form games (Figure 2), for a combination of reasons. First, we observe that the stepsize $\eta$ calculated in Appendix J to make the operator $F$ a contraction in extensive-form games is prohibitively small in the games we test on, each of which has a number of sequences on the order of tens of thousands. At the same time, we observe that ignoring the issue by setting a large constant stepsize in practice often leads to non-convergence of the fixed point iterations. To sidestep both issues, we considered a variant of the algorithm which only performs a single fixed-point iteration, and uses a stepsize hyperparameter $\eta$, where we pick the best from the set $\{1, 10, 20\}$. We remark that this variant of the algorithm is clairvoyant only in spirit, and while it is a sound regret-minimization algorithm, we expect that the strong theoretical

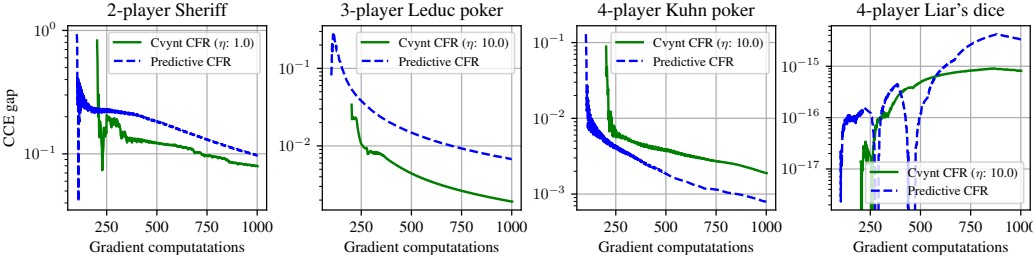

Figure 3: Practical performance of our variant of clairvoyant CFR ('Cvynt CFR') compared to predictive CFR, across four multiplayer extensive-form games. Note that on Liar's dice, both algorithms are down to machine-precision accuracy immediately, which explains the jittery plot.

guarantees of constant per-player regret do not apply. Nevertheless, in Fig. 3 we show that we are able to sometimes observe superior performance to (non-clairvoyant) predictive CFR in the four games we tried, which are described in the appendix. For both algorithms, we ignore the first 100 iterations, in which the iterates are very far from convergence. To compensate for the increased amount of computation needed at each iteration by our clairvoyant algorithm, we plot on the x-axis not the number of iterations but rather the number of gradients of the utility functions computed for each player. On the y-axis, we measure the gap to a coarse correlated equilibrium, which is equal to the maximum regret across the players, divided by the number of iterations.

## 7 Conclusion

We initiated the study of stability for $RM^+$, and showed that both $RM^+$ and predictive $RM^+$ suffer from stability issues that can lead to slow convergence in games. We introduced two simple ideas, *restarting* and *chopping off*, that ensure stability. Consequently, we introduced stable/smooth Predictive $RM^+$, conceptual $RM^+$ and Extragradient $RM^+$, all with strong regret guarantees. Our results yield the first $RM^+$-based algorithms with better than $O(\sqrt{T})$ regret guarantees, thus partially resolving the open question of whether optimism can yield theoretical speedup for $RM^+$. Future directions include understanding whether our stability observations can be leveraged more directly in $RM^+$ without adding our stability tricks, extending our results to general convex games, for which a regret minimizer based on Blackwell approachability similar to $RM^+$ has been proposed recently [17], and combining clairvoyant updates with alternation.

**Funding.** J. Grand-Clément is supported by the Agence Nationale de la Recherche [Grant 11-LABX-0047] and by Hi! Paris. Christian Kroer is supported by the Office of Naval Research awards N00014-22-1-2530 and N00014-23-1-2374, and the National Science Foundation awards IIS-2147361 and IIS-2238960. Haipeng Luo and Chung-Wei Lee are supported by National Science Foundation award IIS-1943607.

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

# A    Proof of Lemma 2.1

*Proof of Lemma 2.1.* Let us write $\hat{\boldsymbol{R}} = \hat{\boldsymbol{x}}$. Note that

$$
\begin{aligned}
\mathrm{Reg}^T(\hat{\boldsymbol{x}}) &= \sum_{t=1}^T \langle \boldsymbol{x}^t, \boldsymbol{\ell}^t \rangle - \sum_{t=1}^T \langle \hat{\boldsymbol{x}}, \boldsymbol{\ell}^t \rangle \\
&= -\sum_{t=1}^T \langle \hat{\boldsymbol{x}}, \boldsymbol{f}(\boldsymbol{x}^t, \boldsymbol{\ell}^t) \rangle && (3) \\
&= -\sum_{t=1}^T \left\langle \hat{\boldsymbol{R}}, \boldsymbol{f}(\boldsymbol{x}^t, \boldsymbol{\ell}^t) \right\rangle \\
&= \sum_{t=1}^T \langle \boldsymbol{R}^t, \boldsymbol{f}(\boldsymbol{x}^t, \boldsymbol{\ell}^t) \rangle - \sum_{t=1}^T \left\langle \hat{\boldsymbol{R}}, \boldsymbol{f}(\boldsymbol{x}^t, \boldsymbol{\ell}^t) \right\rangle && (4) \\
&= \mathrm{Reg}^T \left( \hat{\boldsymbol{R}} \right) && (5)
\end{aligned}
$$

where (3) follows from $\hat{\boldsymbol{x}}^\top \mathbf{1}_d = 1$ and the definition of the map $\boldsymbol{f}(\cdot, \cdot)$, (4) follows from $\langle \boldsymbol{R}^t, \boldsymbol{f}(\boldsymbol{x}^t, \boldsymbol{\ell}^t) \rangle = 0$ because $\boldsymbol{x}^t = \boldsymbol{R}^t / \|\boldsymbol{R}^t\|_1$ (note that this is also trivially true when $\boldsymbol{R}^t = \mathbf{0}$), and (5) follows from the definition of $\mathrm{Reg}^T \left( \hat{\boldsymbol{R}} \right)$. $\qquad\square$

# B    Instability of RM$^+$ and predictive RM$^+$

## B.1    Proof of Theorem 3.1

*Proof of Theorem 3.1.* We first prove the case for RM$^+$. Since we consider $\boldsymbol{x}^t \in \mathbb{R}^2$, we can express $\boldsymbol{x}^t = (p^t, 1 - p^t)$ for some scalar $p^t \in [0, 1]$ (starting with $p^1 = 1/2$). In our counterexample, we set $\boldsymbol{\ell}^t = (\ell^t, 0)$ for some scalar $\ell^t$ to be specified. Consequently, we have

$$
\boldsymbol{f}(\boldsymbol{x}^t, \boldsymbol{\ell}^t) = \boldsymbol{\ell}^t - \langle \boldsymbol{x}^t, \boldsymbol{\ell}^t \rangle \mathbf{1}_2 = ((1 - p^t)\ell^t, \ -p^t \ell^t).
$$

To make the algorithm highly unstable, we first provide an unbounded sequence of $\ell^t$ so that the resulting $\boldsymbol{R}^t$ alternates between vectors with the same value on both entries and vectors with only the first entry being 0, which means $p^t$ by definition alternates between $1/2$ and 0. Noting that RM+ is scale-invariant to the loss sequence, our proof is completed by normalizing the losses so that they all lie in $[-1, 1]$.

Specifically, we set $\ell^1 = 2$, which gives $\boldsymbol{f}(\boldsymbol{x}^1, \boldsymbol{\ell}^1) = (1, -1)$, $\boldsymbol{R}^2 = (0, 1)$, and $p^2 = 0$. Then for $t \geq 2$ we set $\ell^t = -2^{(t-2)/2}$ when $t$ is even and $\ell^t = 2^{(t-1)/2}$ when $t$ is odd. By direct calculation it is not hard to verify that

$$
\boldsymbol{f}(\boldsymbol{x}^t, \boldsymbol{\ell}^t) = (-2^{(t-2)/2}, 0), \qquad\qquad \boldsymbol{R}^{t+1} = (2^{(t-2)/2}, 2^{(t-2)/2}),
$$

$p^{t+1} = \frac{1}{2}$ when $t$ is even, and

$$
\boldsymbol{f}(\boldsymbol{x}^t, \boldsymbol{\ell}^t) = (2^{(t-3)/2}, -2^{(t-3)/2}), \qquad\qquad \boldsymbol{R}^{t+1} = (0, 2^{(t-1)/2}),
$$

$p^{t+1} = 0$ when $t$ is odd, completing the counterexample for RM$^+$.

It remains to prove the case for predictive RM$^+$, where $\boldsymbol{m}^t = \boldsymbol{f}(\boldsymbol{x}^{t-1}, \boldsymbol{\ell}^{t-1})$. Initially, let $\ell^1 = 4$, $\ell^2 = -1$. Recall that

$$
\boldsymbol{f}(\boldsymbol{x}^t, \boldsymbol{\ell}^t) = \boldsymbol{\ell}^t - \langle \boldsymbol{x}^t, \boldsymbol{\ell}^t \rangle \mathbf{1}_2 = ((1 - p^t)\ell^t, \ -p^t \ell^t).
$$

By direct calculation, we have

$$
\begin{aligned}
\boldsymbol{f}(\boldsymbol{x}^1, \boldsymbol{\ell}^1) = (2, -2), &\qquad \boldsymbol{R}^2 = (0, 2), &\qquad \hat{\boldsymbol{R}}^2 = (-2, 4), &\qquad p^2 = 0 \\
\boldsymbol{f}(\boldsymbol{x}^2, \boldsymbol{\ell}^2) = (-1, 0), &\qquad \boldsymbol{R}^3 = (1, 2), &\qquad \hat{\boldsymbol{R}}^3 = (2, 2), &\qquad p^3 = \frac{1}{2}
\end{aligned}
$$

Thereafter, we set $\ell^t = 2^{(t+1)/2}$ when $t$ is odd and $\ell^t = -2^{(t-2)/2}$ when $t$ is even. The updates for the next 4 steps are:

$$\boldsymbol{f}(\boldsymbol{x}^3, \boldsymbol{\ell}^3) = (2, -2), \qquad \boldsymbol{R}^4 = (0, 4), \qquad \hat{\boldsymbol{R}}^4 = (0, 6), \qquad p^4 = 0$$

$$\boldsymbol{f}(\boldsymbol{x}^4, \boldsymbol{\ell}^4) = (-2, 0), \qquad \boldsymbol{R}^5 = (2, 4), \qquad \hat{\boldsymbol{R}}^5 = (4, 4), \qquad p^5 = \frac{1}{2}$$

$$\boldsymbol{f}(\boldsymbol{x}^5, \boldsymbol{\ell}^5) = (4, -4), \qquad \boldsymbol{R}^6 = (0, 8), \qquad \hat{\boldsymbol{R}}^6 = (0, 12), \qquad p^6 = 0$$

$$\boldsymbol{f}(\boldsymbol{x}^6, \boldsymbol{\ell}^6) = (-4, 0), \qquad \boldsymbol{R}^7 = (4, 8), \qquad \hat{\boldsymbol{R}}^7 = (8, 8), \qquad p^7 = \frac{1}{2}.$$

It is not hard to verify (by induction) that

$$\boldsymbol{f}(\boldsymbol{x}^t, \boldsymbol{\ell}^t) = (2^{(t-1)/2}, -2^{(t-1)/2}), \quad \boldsymbol{R}^{t+1} = (0, 2^{(t+1)/2}), \quad \hat{\boldsymbol{R}}^{t+1} = (0, 2^{(t+1)/2} + 2^{(t-1)/2}), \quad p^{t+1} = 0$$

when $t$ is odd and

$$\boldsymbol{f}(\boldsymbol{x}^t, \boldsymbol{\ell}^t) = (-2^{(t-2)/2}, 0), \quad \boldsymbol{R}^{t+1} = (2^{(t-2)/2}, 2^{t/2}), \quad \hat{\boldsymbol{R}}^{t+1} = (2^{t/2}, 2^{t/2}), \quad p^{t+1} = \frac{1}{2}$$

when $t$ is even. This completes the proof. $\qquad\square$

**Remark B.1.** *The losses are unbounded in the examples, but note that the update rules for the algorithms imply that all the algorithms remain unchanged after scaling the losses, so we can rescale them accordingly. Specifically, if we have a loss sequence $\ell^1, \dots, \ell^T$, we can define $L_T = \max\{|\ell^1|, \dots, |\ell^T|\}$ and consider another loss sequence $\ell^1/L_T, \dots, \ell^T/L_T$, which is bounded in $[-1, 1]$ and will make the algorithms produce the same outputs.*

### B.2 Counterexample on $3 \times 3$ matrix game for RM$^+$

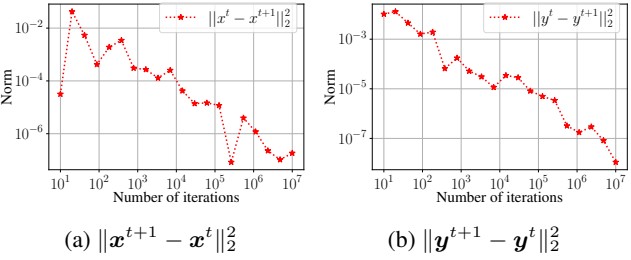

(a) $\|\boldsymbol{x}^{t+1} - \boldsymbol{x}^t\|_2^2$      (b) $\|\boldsymbol{y}^{t+1} - \boldsymbol{y}^t\|_2^2$

Figure 4: $\|\boldsymbol{x}^{t+1} - \boldsymbol{x}^t\|_2^2$ (Figure 4a) and $\|\boldsymbol{y}^{t+1} - \boldsymbol{y}^t\|_2^2$ (Figure 4b) for RM$^+$.

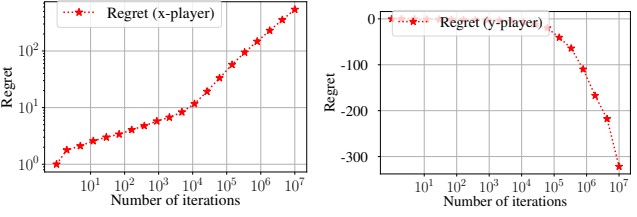

(a) Regret of the first player      (b) Regret of the second player

Figure 5: Individual regret of each player for RM$^+$.

## C  Proof of Proposition 1 and Corollary 3.2

We start with a couple of technical lemmas.

**Lemma C.1.** *Given any $\boldsymbol{y} \in \mathbb{R}_+^d$ and $\boldsymbol{x} \in \mathbb{R}^d$ such that $\mathbf{1}^\top \boldsymbol{x} = 0$,*

$$(\boldsymbol{x}^\top \boldsymbol{y})^2 \le \frac{d-1}{d} \|\boldsymbol{x}\|_2^2 \|\boldsymbol{y}\|_2^2.$$

*Proof.* If $\boldsymbol{x} = \boldsymbol{0}$ the claim is trivial, so we focus on the other case. Let $\boldsymbol{\xi}$ be the (Euclidean) projection of $\boldsymbol{y}$ onto the orthogonal complement of $\mathrm{span}\{\boldsymbol{x}, \boldsymbol{1}\}$. Since by hypothesis $\boldsymbol{1} \perp \boldsymbol{x}$, it holds that

$$\boldsymbol{y} = \frac{\boldsymbol{y}^\top \boldsymbol{x}}{\|\boldsymbol{x}\|_2^2} \boldsymbol{x} + \frac{\boldsymbol{1}^\top \boldsymbol{y}}{\|\boldsymbol{1}\|_2^2} \boldsymbol{1} + \boldsymbol{\xi}$$

and therefore

$$\|\boldsymbol{y}\|_2^2 = \frac{(\boldsymbol{y}^\top \boldsymbol{x})^2}{\|\boldsymbol{x}\|_2^2} + \frac{(\boldsymbol{1}^\top \boldsymbol{y})^2}{\|\boldsymbol{1}\|_2^2} + \|\boldsymbol{\xi}\|_2^2 \geq \frac{(\boldsymbol{y}^\top \boldsymbol{x})^2}{\|\boldsymbol{x}\|_2^2} + \frac{(\boldsymbol{1}^\top \boldsymbol{y})^2}{\|\boldsymbol{1}\|_2^2} \tag{6}$$

Using the hypothesis that $\boldsymbol{y} \geq \boldsymbol{0}$, we can bound

$$\boldsymbol{1}^\top \boldsymbol{y} = \|\boldsymbol{y}\|_1 \geq \|\boldsymbol{y}\|_2,$$

where we used the well-known inequality between the $\ell_1$-norm and the $\ell_2$-norm. Substituting the previous inequality into (6), and using the fact that $\|\boldsymbol{1}\|_2^2 = d$,

$$\|\boldsymbol{y}\|_2^2 \geq \frac{(\boldsymbol{y}^\top \boldsymbol{x})^2}{\|\boldsymbol{x}\|_2^2} + \frac{\|\boldsymbol{y}\|_2^2}{d}.$$

Rearranging the terms yields the statement. $\qquad\square$

**Lemma C.2.** *For any $\hat{\boldsymbol{y}} \in \mathbb{R}_+^d$ such that $\|\hat{\boldsymbol{y}}\|_2 = 1$, $\boldsymbol{1}^\top \hat{\boldsymbol{y}} \neq 0$ and for any $\boldsymbol{x} \in \mathbb{R}^d$ such that $\boldsymbol{1}^\top \boldsymbol{x} = 1$,*

$$\left( \frac{1}{\boldsymbol{1}^\top \hat{\boldsymbol{y}}} - \boldsymbol{x}^\top \hat{\boldsymbol{y}} \right)^2 \leq (d-1) \cdot \left\| (\boldsymbol{x}^\top \hat{\boldsymbol{y}}) \hat{\boldsymbol{y}} - \boldsymbol{x} \right\|_2^2.$$

*Proof.* The main idea of the proof is to introduce

$$\boldsymbol{z} := \boldsymbol{x} - \frac{\hat{\boldsymbol{y}}}{\boldsymbol{1}^\top \hat{\boldsymbol{y}}}.$$

Note that $\boldsymbol{1}^\top \boldsymbol{z} = \boldsymbol{1}^\top \boldsymbol{x} - 1 = 0$. Furthermore,

$$\boldsymbol{x}^\top \hat{\boldsymbol{y}} = \left( \boldsymbol{z} + \frac{\hat{\boldsymbol{y}}}{\boldsymbol{1}^\top \hat{\boldsymbol{y}}} \right)^\top \hat{\boldsymbol{y}} = \boldsymbol{z}^\top \hat{\boldsymbol{y}} + \frac{1}{\boldsymbol{1}^\top \hat{\boldsymbol{y}}}.$$

Substituting the previous equality in the statement, we obtain

$$\left( \frac{1}{\boldsymbol{1}^\top \hat{\boldsymbol{y}}} - \boldsymbol{x}^\top \hat{\boldsymbol{y}} \right)^2 - (d-1) \cdot \left\| (\boldsymbol{x}^\top \hat{\boldsymbol{y}}) \hat{\boldsymbol{y}} - \boldsymbol{x} \right\|_2^2$$

$$= (\boldsymbol{z}^\top \hat{\boldsymbol{y}})^2 - (d-1) \cdot \left\| \left( \boldsymbol{z}^\top \hat{\boldsymbol{y}} + \frac{1}{\boldsymbol{1}^\top \hat{\boldsymbol{y}}} \right) \hat{\boldsymbol{y}} - \boldsymbol{z} - \frac{\hat{\boldsymbol{y}}}{\boldsymbol{1}^\top \hat{\boldsymbol{y}}} \right\|_2^2$$

$$= (\boldsymbol{z}^\top \hat{\boldsymbol{y}})^2 - (d-1) \cdot \left\| (\boldsymbol{z}^\top \hat{\boldsymbol{y}}) \hat{\boldsymbol{y}} - \boldsymbol{z} \right\|_2^2$$

$$= (\boldsymbol{z}^\top \hat{\boldsymbol{y}})^2 - (d-1) \left( (\boldsymbol{z}^\top \hat{\boldsymbol{y}})^2 + \|\boldsymbol{z}\|_2^2 - 2(\boldsymbol{z}^\top \hat{\boldsymbol{y}})^2 \right)$$

$$= d \left( (\boldsymbol{z}^\top \hat{\boldsymbol{y}})^2 - \frac{d-1}{d} \|\boldsymbol{z}\|_2^2 \right),$$

where we used the hypothesis that $\|\hat{\boldsymbol{y}}\|_2^2 = 1$ in the third equality. Using the inequality of Lemma C.1 concludes the proof. $\qquad\square$

We are now ready to prove Proposition 1.

*Proof of Proposition 1.* If $\boldsymbol{y} = \boldsymbol{0}$, the statement holds trivially. Hence, we focus on the case $\boldsymbol{y} \neq \boldsymbol{0}$. Let $\hat{\boldsymbol{y}} := \boldsymbol{y} / \|\boldsymbol{y}\|_2$ be the direction of $\boldsymbol{y}$; clearly, $\|\hat{\boldsymbol{y}}\|_2 = 1$. Note that

$$\|\boldsymbol{y} - \boldsymbol{x}\|_2^2 = \left( \|\boldsymbol{y}\|_2 - \boldsymbol{x}^\top \hat{\boldsymbol{y}} \right)^2 + \left\| \boldsymbol{x} - (\boldsymbol{x}^\top \hat{\boldsymbol{y}}) \hat{\boldsymbol{y}} \right\|_2^2$$

$$\geq \left\| \boldsymbol{x} - (\boldsymbol{x}^\top \hat{\boldsymbol{y}}) \hat{\boldsymbol{y}} \right\|_2^2$$

$$= (\mathbf{1}^\top \boldsymbol{x})^2 \left\| \boldsymbol{g}(\boldsymbol{x}) - \left( \boldsymbol{g}(\boldsymbol{x})^\top \hat{\boldsymbol{y}} \right) \hat{\boldsymbol{y}} \right\|_2^2$$

$$\geq \left\| \boldsymbol{g}(\boldsymbol{x}) - \left( \boldsymbol{g}(\boldsymbol{x})^\top \hat{\boldsymbol{y}} \right) \hat{\boldsymbol{y}} \right\|_2^2, \tag{7}$$

where we used the hypothesis that $\mathbf{1}^\top \boldsymbol{x} \geq 1$ in the last step. On the other hand, using Lemma C.2 (note that $\mathbf{1}^\top \hat{\boldsymbol{y}} \neq 0$ since $\boldsymbol{y} \neq 0$ by hypothesis),

$$\left\| \boldsymbol{g}(\boldsymbol{x}) - \left( \boldsymbol{g}(\boldsymbol{x})^\top \hat{\boldsymbol{y}} \right) \hat{\boldsymbol{y}} \right\|_2^2$$

$$= \frac{1}{d} \left\| \boldsymbol{g}(\boldsymbol{x}) - \left( \boldsymbol{g}(\boldsymbol{x})^\top \hat{\boldsymbol{y}} \right) \hat{\boldsymbol{y}} \right\|_2^2$$

$$+ \frac{d-1}{d} \left\| \boldsymbol{g}(\boldsymbol{x}) - \left( \boldsymbol{g}(\boldsymbol{x})^\top \hat{\boldsymbol{y}} \right) \hat{\boldsymbol{y}} \right\|_2^2$$

$$\geq \frac{1}{d} \left\| \boldsymbol{g}(\boldsymbol{x}) - \left( \boldsymbol{g}(\boldsymbol{x})^\top \hat{\boldsymbol{y}} \right) \hat{\boldsymbol{y}} \right\|_2^2 + \frac{1}{d} \left( \frac{1}{\mathbf{1}^\top \hat{\boldsymbol{y}}} - \boldsymbol{g}(\boldsymbol{x})^\top \hat{\boldsymbol{y}} \right)^2$$

$$= \frac{1}{d} \left( \left\| \boldsymbol{g}(\boldsymbol{x}) \right\|_2^2 + \frac{1}{(\mathbf{1}^\top \hat{\boldsymbol{y}})^2} - 2 \frac{\boldsymbol{g}(\boldsymbol{x})^\top \hat{\boldsymbol{y}}}{\mathbf{1}^\top \hat{\boldsymbol{y}}} \right)$$

$$= \frac{1}{d} \left( \left\| \boldsymbol{g}(\boldsymbol{x}) \right\|_2^2 + \left\| \boldsymbol{g}(\boldsymbol{y}) \right\|_2^2 - 2 \boldsymbol{g}(\boldsymbol{x})^\top \boldsymbol{g}(\boldsymbol{y}) \right)$$

$$= \frac{1}{d} \left\| \boldsymbol{g}(\boldsymbol{y}) - \boldsymbol{g}(\boldsymbol{x}) \right\|_2^2. \tag{8}$$

Combining (7) and (8), we obtain the statement. $\qquad \square$

*Proof of Corollary 3.2.* The condition means that $\mathbf{1}^\top \frac{\boldsymbol{R}^t}{R_0} \geq 1$ and

$$\left\| \boldsymbol{x}^{t+1} - \boldsymbol{x}^t \right\|_2 \leq \sqrt{d} \left\| \frac{\boldsymbol{R}^{t+1}}{R_0} - \frac{\boldsymbol{R}^t}{R_0} \right\|_2 \qquad \text{(by Proposition 1)}$$

$$\leq \frac{\sqrt{d}}{R_0} \left\| \boldsymbol{f}(\boldsymbol{x}^t, \boldsymbol{\ell}^t) \right\|_2$$

$$\leq \frac{\sqrt{d}}{R_0} \left( \left\| \boldsymbol{\ell}^t \right\|_2 + \left\| \langle \boldsymbol{x}^t, \boldsymbol{\ell}^t \rangle \mathbf{1}_d \right\|_2 \right)$$

$$\leq \frac{\sqrt{d}}{R_0} \left( B_u + \sqrt{d} \left\| \boldsymbol{x}^t \right\|_2 \left\| \boldsymbol{\ell}^t \right\|_2 \right) \qquad \text{(by (1))}$$

$$\leq \frac{2 d B_u}{R_0}$$

$\qquad \square$

# D  Proof of Theorem 4.1

*Proof of Theorem 4.1.* When the algorithm restarts, the accumulated regret is negative to all actions, so it is sufficient to consider the regret from $T_0$, the round when the last restart happens to the end. In that case, we can analyze the algorithm as a normal predictive regret matching algorithm. By Proposition 5 in [12], we have that the regret for player $i$ is bounded by

$$\mathrm{Reg}_i^T(\boldsymbol{x}^*) \leq \frac{\left\| \boldsymbol{x}^* - \boldsymbol{z}_i^{T_0} \right\|_2^2}{2\eta} + \eta \sum_{t=T_0}^T \left\| \boldsymbol{f}_i^t - \boldsymbol{m}_i^t \right\|^2 - \frac{1}{8\eta} \sum_{T=T_0}^T \left\| \boldsymbol{z}_i^{t+1} - \boldsymbol{z}_i^t \right\|^2, \tag{9}$$

where $\boldsymbol{z}_i^{T_0} = (R_0, \dots, R_0)$ and $\left( \boldsymbol{f}_i^{t-1} \right)_{i \in [n]} = F(\boldsymbol{z}^{t-1})$. When setting $\boldsymbol{m}_i^t = \boldsymbol{f}_i^{t-1}$, then $\left\| \boldsymbol{f}_i^t - \boldsymbol{m}_i^t \right\|$ can be bounded by

$$\left\| \boldsymbol{f}_i^t - \boldsymbol{m}_i^t \right\|_2 = \left\| \langle \boldsymbol{x}_i^t, \boldsymbol{\ell}_i^t \rangle \mathbf{1}_{d_i} - \langle \boldsymbol{x}_i^{t-1}, \boldsymbol{\ell}_i^{t-1} \rangle \mathbf{1}_{d_i} - \left( \boldsymbol{\ell}_i^t - \boldsymbol{\ell}_i^{t-1} \right) \right\|_2$$

$$
\begin{aligned}
&= \left\| \left\langle \boldsymbol{x}_i^t - \boldsymbol{x}_i^{t-1}, \boldsymbol{\ell}_i^t \right\rangle \mathbf{1}_{d_i} - \left\langle \boldsymbol{x}_i^{t-1}, \boldsymbol{\ell}_i^{t-1} - \boldsymbol{\ell}_i^t \right\rangle \mathbf{1}_{d_i} - \left( \boldsymbol{\ell}_i^t - \boldsymbol{\ell}_i^{t-1} \right) \right\|_2 \\
&\le \left\| \boldsymbol{x}_i^t - \boldsymbol{x}_i^{t-1} \right\|_2 \left\| \boldsymbol{\ell}_i^t \right\|_2 \|\mathbf{1}_{d_i}\|_2 + \left\| \boldsymbol{x}_i^{t-1} \right\|_2 \left\| \boldsymbol{\ell}_i^{t-1} - \boldsymbol{\ell}_i^t \right\|_2 \|\mathbf{1}_{d_i}\|_2 + \left\| \boldsymbol{\ell}_i^t - \boldsymbol{\ell}_i^{t-1} \right\|_2 \\
&\le B_u \sqrt{d_i} \left\| \boldsymbol{x}_i^t - \boldsymbol{x}_i^{t-1} \right\|_2 + \sqrt{d_i} \left\| \boldsymbol{\ell}_i^t - \boldsymbol{\ell}_i^{t-1} \right\|_2 + \left\| \boldsymbol{\ell}_i^t - \boldsymbol{\ell}_i^{t-1} \right\|_2 && \text{(by (1))} \\
&\le B_u \sqrt{d_i} \left\| \boldsymbol{x}_i^t - \boldsymbol{x}_i^{t-1} \right\|_2 + \sqrt{d_i} \sum_{i' \ne i} 2 L_u \left\| \boldsymbol{x}_{i'}^t - \boldsymbol{x}_{i'}^{t-1} \right\|_2 && \text{(by (1))} \\
&\le 2 \sqrt{d_i} (B_u + L_u) \sum_{i' \in [n]} \left\| \boldsymbol{x}_{i'}^t - \boldsymbol{x}_{i'}^{t-1} \right\|_2 \\
&\le \frac{12 \eta \sqrt{d_i} B_u (B_u + L_u) \sum_{i' \in [n]} d_{i'}}{R_0} \\
&\le \frac{12 \eta B_u (B_u + L_u) d^{3/2}}{R_0}
\end{aligned}
$$

where the last-but-one inequality is because

$$
\begin{aligned}
\left\| \boldsymbol{x}_i^t - \boldsymbol{x}_i^{t-1} \right\|_2 &= \left\| \boldsymbol{g}(\boldsymbol{z}_i^t) - \boldsymbol{g}(\boldsymbol{z}_i^{t-1}) \right\|_2 \\
&\le \left\| \boldsymbol{g}(\boldsymbol{z}_i^t) - \boldsymbol{g}(\boldsymbol{w}_i^{t-1}) \right\|_2 + \left\| \boldsymbol{g}(\boldsymbol{w}_i^{t-1}) - \boldsymbol{g}(\boldsymbol{w}_i^{t-2}) \right\|_2 + \left\| \boldsymbol{g}(\boldsymbol{w}_i^{t-2}) - \boldsymbol{g}(\boldsymbol{z}_i^{t-1}) \right\|_2 \\
&\le 3 \cdot \frac{2 \eta d_i B_u}{R_0} = \frac{6 \eta d_i B_u}{R_0}
\end{aligned}
$$

and we bound each of RHS of the first inequality using a restatement of Corollary 3.2, shown in Lemma D.1. Therefore, we can further bound (9) it by dropping the negative terms and bounding the rest by

$$
\frac{\|\boldsymbol{x}^*\|_2^2 + \|\boldsymbol{z}_i^{T_0}\|_2^2}{\eta} + \eta \sum_{t=T_0}^{T} \left\| \boldsymbol{f}_i^t - \boldsymbol{f}_i^{t-1} \right\|^2 \le \frac{1 + R_0^2 d}{\eta} + \eta^3 T \cdot \frac{144 B_u^2 (B_u + L_u)^2 d^3}{R_0^2}.
$$

Choosing $R_0 = 1$ and $\eta = \left( d^2 T \right)^{-1/4}$ finishes the proof. $\qquad \square$

**Lemma D.1.** *Let* $\boldsymbol{z} = \Pi_{\boldsymbol{w}, \mathcal{X}} \left( \eta \boldsymbol{f}(\boldsymbol{x}, \boldsymbol{\ell}) \right)$ *for* $\boldsymbol{x} \in \Delta^d$, $\boldsymbol{\ell} \in \mathbb{R}^d$, $\|\boldsymbol{\ell}\|_2 \le B_u$. *Suppose* $\|\boldsymbol{w}\|_1 \ge R_0$, *then we have*

$$
\|\boldsymbol{g}(\boldsymbol{z}) - \boldsymbol{g}(\boldsymbol{w})\|_2 \le \frac{\sqrt{d}}{R_0} \cdot \|\boldsymbol{z} - \boldsymbol{w}\|_2 \le \frac{2 \eta d B_u}{R_0}.
$$

*Proof.* The proof is essentially the same as Corollary 3.2. The condition means that $\mathbf{1}^\top \frac{\boldsymbol{w}}{R_0} \ge 1$ and

$$
\begin{aligned}
\|\boldsymbol{g}(\boldsymbol{z}) - \boldsymbol{g}(\boldsymbol{w})\|_2 &= \|\boldsymbol{g}(\boldsymbol{z}/R_0) - \boldsymbol{g}(\boldsymbol{w}/R_0)\|_2 \\
&\le \sqrt{d} \left\| \frac{\boldsymbol{z}}{R_0} - \frac{\boldsymbol{w}}{R_0} \right\|_2 && \text{(by Proposition 1)} \\
&\le \frac{\sqrt{d}}{R_0} \|\eta \boldsymbol{f}(\boldsymbol{x}, \boldsymbol{\ell})\|_2 \\
&\le \frac{\eta \sqrt{d}}{R_0} \left( \|\boldsymbol{\ell}\|_2 + \|\langle \boldsymbol{x}, \boldsymbol{\ell} \rangle \mathbf{1}_d\|_2 \right) \\
&\le \frac{\eta \sqrt{d}}{R_0} \left( B_u + \sqrt{d} \|\boldsymbol{x}\|_2 \|\boldsymbol{\ell}\|_2 \right) && \text{(by (1))} \\
&\le \frac{2 \eta d B_u}{R_0}.
\end{aligned}
$$

$\qquad \square$

# E   Proof of Theorem 4.2

We write $(\boldsymbol{R}_1^t, ..., \boldsymbol{R}_n^t) = \boldsymbol{z}^t$. Let us consider the regret $\mathrm{Reg}_i^T(\hat{\boldsymbol{x}}_i)$ of player $i \in \{1, ..., n\}$. Lemma 2.1 shows that

$$\mathrm{Reg}_i^T(\hat{\boldsymbol{x}}_i) = \mathrm{Reg}_i^T\left(\hat{\boldsymbol{R}}_i\right)$$

with $\mathrm{Reg}_i^T\left(\hat{\boldsymbol{R}}_i\right)$ the regret against $\hat{\boldsymbol{R}}_i = \hat{\boldsymbol{x}}_i$ incurred by a decision-maker choosing the decisions $(\boldsymbol{R}_i^t)_{t\geq 1}$ and facing the sequence of losses $(\boldsymbol{f}_i^t)_{t\geq 1}$, with $\boldsymbol{f}_i^t = \boldsymbol{\ell}_i^t - \langle \boldsymbol{x}_i^t, \boldsymbol{\ell}_i^t \rangle \mathbf{1}_{d_i}$:

$$\mathrm{Reg}_i^T\left(\hat{\boldsymbol{R}}_i\right) = \sum_{t=1}^T \left\langle \boldsymbol{f}_i^t, \boldsymbol{R}_i^t - \hat{\boldsymbol{R}}_i \right\rangle \tag{10}$$

Note that $\boldsymbol{R}_i^1, ..., \boldsymbol{R}_i^T$ is computed by Predictive OMD with $\Delta_{\geq}^{d_i}$ as a decision set, $\boldsymbol{f}_i^1, ..., \boldsymbol{f}_i^T$ as the sequence of losses and $\boldsymbol{m}_i^1, ..., \boldsymbol{m}_i^T$ as the sequence of predictions. Therefore, Proposition 5 in [12] applies, and we can write the following regret bound:

$$\begin{aligned}
\mathrm{Reg}_i^T\left(\hat{\boldsymbol{R}}_i\right) \leq{}& \frac{\|\boldsymbol{w}_i^0 - \hat{\boldsymbol{x}}_i\|_2^2}{2\eta} + \eta \sum_{t=1}^T \left\|\boldsymbol{f}_i^t - \boldsymbol{m}_i^t\right\|_2^2 \\
& - \frac{1}{8\eta} \sum_{t=1}^T \left\|\boldsymbol{R}_i^{t+1} - \boldsymbol{R}_i^t\right\|_2^2.
\end{aligned} \tag{11}$$

Since we maintain $\boldsymbol{R}_i^t \in \Delta_{\geq}^{d_i}$ at every iteration, using Proposition 1 we find that

$$\|\boldsymbol{x}_i^{t+1} - \boldsymbol{x}_i^t\|_2^2 \leq d_i \left\|\boldsymbol{R}_i^{t+1} - \boldsymbol{R}_i^t\right\|_2^2.$$

Plugging this into (11), we obtain

$$\begin{aligned}
\mathrm{Reg}_i^T\left(\hat{\boldsymbol{R}}_i\right) \leq{}& \frac{\|\boldsymbol{w}_i^0 - \hat{\boldsymbol{x}}_i\|_2^2}{2\eta} + \eta \sum_{t=1}^T \left\|\boldsymbol{f}_i^t - \boldsymbol{m}_i^t\right\|_2^2 \\
& - \frac{1}{8d_i\eta} \sum_{t=1}^T \left\|\boldsymbol{x}^{t+1} - \boldsymbol{x}^t\right\|_2^2.
\end{aligned}$$

Using $\|\cdot\|_2 \leq \|\cdot\|_1 \leq \sqrt{d_i}\|\cdot\|_2$, we obtain

$$\begin{aligned}
\mathrm{Reg}_i^T\left(\hat{\boldsymbol{R}}_i\right) \leq{}& \alpha + \beta \sum_{t=1}^T \left\|\boldsymbol{f}_i^t - \boldsymbol{m}_i^t\right\|_1^2 \\
& - \gamma \sum_{t=1}^T \left\|\boldsymbol{x}^{t+1} - \boldsymbol{x}^t\right\|_1^2.
\end{aligned}$$

with $\alpha = \frac{\|\boldsymbol{w}_i^0 - \hat{\boldsymbol{x}}_i\|_2^2}{2\eta}, \beta = d_i\eta, \gamma = \frac{1}{8d_i^2\eta}$. To conclude as in Theorem 4 in [34] we need $\beta \leq \gamma/(n-1)^2$, i.e., $\eta = \frac{1}{2\sqrt{2}(n-1)d_i^{3/2}}$. Therefore, using $\eta = \frac{1}{2\sqrt{2}(n-1)\max_i\{d_i^{3/2}\}}$, we conclude that the sum of the individual regrets is bounded by

$$O\left(n^2 \max_{i=1,...,n}\{d_i^{3/2}\} \max_{i=1,...,n}\{\|\boldsymbol{w}_i^0 - \hat{\boldsymbol{x}}_i\|_2^2\}\right).$$

# F   Proof of Theorem 5.3

*Proof of Lemma 5.1.* The first-order optimality condition gives

$$\langle \eta\boldsymbol{\ell}^t + \boldsymbol{z}^t - \boldsymbol{z}^{t-1}, \hat{\boldsymbol{z}} - \boldsymbol{z}^t \rangle \geq 0 \ \forall \hat{\boldsymbol{z}} \in \mathcal{Z}.$$

Rearranging gives that for any $\hat{z} \in \mathcal{Z}$, we have

$$\langle \eta \ell^t, \hat{z} - z^t \rangle \geq \langle z^{t-1} - z^t, \hat{z} - z^t \rangle = \frac{1}{2} \|z^t - \hat{z}\|_2^2 - \frac{1}{2} \|z^{t-1} - \hat{z}\|_2^2 + \frac{1}{2} \|z^t - z^{t-1}\|_2^2.$$

Multiplying by $-1$ and summing over all $t = 1, ..., T$ gives the regret bound:

$$\sum_{t=1}^{T} \langle \eta \ell^t, z^t - \hat{z} \rangle \leq \frac{1}{2} \|z^0 - \hat{z}\|_2^2 - \frac{1}{2} \|z^T - \hat{z}\|_2^2 - \sum_{t=1}^{T} \frac{1}{2} \|z^t - z^{t-1}\|_2^2 \leq \frac{1}{2} \|z^0 - \hat{z}\|_2^2.$$

$\square$

*Proof of Lemma 5.2.* Let $x, x' \in \Delta$ and $i \in \{1, ..., n\}$. Let us write $\ell_i = -\nabla_{x_i} u_i(x), \ell_i' = -\nabla_{x_i} u_i(x')$. We have, for $i \in \{1, ..., n\}$,

$$\begin{aligned}
\|\boldsymbol{f}\left(x_i, \ell_i\right) &- \boldsymbol{f}\left(x_i', \ell_i'\right)\|_2^2 \\
&= \sum_{j=1}^{d_i} \left((x_i - e_j)^\top \ell_i - (x_i' - e_j)^\top \ell_i'\right)^2 \\
&= \sum_{j=1}^{d_i} ((x_i - e_j)^\top \ell_i - (x_i' - e_j)^\top \ell_i + (x_i' - e_j)^\top \ell_i - (x_i' - e_j)^\top \ell_i')^2 \\
&= \sum_{j=1}^{d_i} \left((x_i - x_i')^\top \ell_i + (x_i' - e_j)^\top (\ell_i - \ell_i')\right)^2 \\
&\leq 2 d_i \left((x_i - x_i')^\top \ell_i\right)^2 + 2 \sum_{j=1}^{d_i} \left((x_i' - e_j)^\top (\ell_i - \ell_i')\right)^2 \\
&\leq 2 d_i \|x_i - x_i'\|_2^2 \|\ell_i\|_2^2 + 2 \sum_{j=1}^{d_i} \|x_i' - e_j\|_2^2 \|\ell_i - \ell_i'\|_2^2,
\end{aligned}$$

where the last inequality follows from Cauchy-Schwarz inequality. Now from (1) and the definition of $\ell_i, \ell_i'$, we have

$$\|\ell_i\|_2 \leq B_u, \|\ell_i - \ell_i'\|_2 \leq L_u \|x - x'\|_2.$$

This yields

$$\begin{aligned}
\|\boldsymbol{f}\left(x_i, \ell_i\right) - \boldsymbol{f}\left(x_i', \ell_i'\right)\|_2^2 &\leq 2 d_i B_u^2 \|x_i - x_i'\|_2^2 + 4 d_i L_u^2 \|x - x'\|_2^2 \\
&\leq \left(2 d_i B_u^2 + 4 d_i L_u^2\right) \|x - x'\|_2^2.
\end{aligned}$$

Since the function $\boldsymbol{g}$ is $\sqrt{d_i}$-Lipschitz continuous over each decision set $\Delta_{\geq}^{d_i}$ (Proposition 1), we have shown that the Lipschitz constant of $F$ is $L_F = (\max_i d_i) \sqrt{2 B_u^2 + 4 L_u^2}$. $\square$

We are now ready to prove Theorem 5.3. We write $(R_1^t, ..., R_n^t) = z^t$.

*Proof of Theorem 5.3.* We use the fact that $(z^t)_{t \geq 1}$ is computed following the Cheating OMD update with $\ell^t = F(z^t)$ at every iteration $t \geq 1$. Therefore, the first-order optimality condition in $z^t = \Pi_{z^{t-1}, \mathcal{X}_{\geq}} (\eta F(z^t))$ yields

$$\langle \eta F(z^t) + z^t - z^{t-1}, \hat{z} - z^t \rangle \geq 0 \, \forall \hat{z} \in \mathcal{X}_{\geq}.$$

Similarly as for the proof of Lemma 5.1, we obtain that for any $\hat{z} \in \mathcal{X}_{\geq}$, we have

$$\langle \eta F(z^t), \hat{z} - z^t \rangle \geq \frac{1}{2} \|z^t - \hat{z}\|_2^2 - \frac{1}{2} \|z^{t-1} - \hat{z}\|_2^2 + \frac{1}{2} \|z^t - z^{t-1}\|_2^2.$$

Let $i \in \{1, ..., n\}$. We apply the inequality above to the vector $\hat{z} \in \mathcal{X}_{\geq} = \times_{i=1}^n \Delta_{\geq}^{d_i}$ defined as $\hat{z}_j = z_j^t$ for $j \neq i$ and $\hat{z}_i = \hat{R}_i$ for some $\hat{R}_i \in \Delta_{\geq}^{d_i}$. This yields, for any $\hat{R}_i \in \Delta_{\geq}^{d_i}$, for $x_j^t = \boldsymbol{g}(R_j^t)$ and $(\ell_1^t, ..., \ell_n^t) = G(x^t)$ for all $j \in \{1, ..., n\}$,

$$\langle \eta \boldsymbol{f}(x^t, \ell^t), \hat{R}_i - R_i^t \rangle \geq \frac{1}{2} \|R_i^t - \hat{R}_i\|_2^2 - \frac{1}{2} \|R_i^{t-1} - \hat{R}_i\|_2^2 + \frac{1}{2} \|R_i^t - R_i^{t-1}\|_2^2.$$

Summing the above inequality for $t = 1, ..., T$, we obtain our bound on the individual regrets of each player: for any $\hat{\boldsymbol{R}}_i \in \Delta^{d_i}_{\geq}$, we have

$$\sum_{t=1}^{T}\langle \eta \boldsymbol{f}(\boldsymbol{x}^t, \boldsymbol{\ell}^t), \boldsymbol{R}_i^t - \hat{\boldsymbol{R}}_i\rangle \leq \frac{1}{2}\|\boldsymbol{R}_i^0 - \hat{\boldsymbol{R}}_i\|_2^2 - \frac{1}{2}\|\boldsymbol{R}_i^T - \hat{\boldsymbol{R}}_i\|_2^2 - \sum_{t=1}^{T}\frac{1}{2}\|\boldsymbol{R}_i^t - \boldsymbol{R}_i^{t-1}\|_2^2 \leq \frac{1}{2}\|\boldsymbol{R}_i^0 - \hat{\boldsymbol{R}}_i\|_2^2.$$

Note that from Lemma 2.1, we have that the individual regret of player $i$

$$\text{Reg}_i^T(\hat{\boldsymbol{x}}_i) = \sum_{t=1}^{T}\left\langle \nabla_{\boldsymbol{x}_i} u_i^t(\boldsymbol{x}^t), \hat{\boldsymbol{x}}_i - \boldsymbol{x}_i^t\right\rangle$$

against a decision $\hat{\boldsymbol{x}}_i \in \Delta^{d_i}$ is equal to $\sum_{t=1}^{T}\langle \boldsymbol{f}(\boldsymbol{x}^t, \boldsymbol{\ell}^t), \boldsymbol{R}_i^t - \hat{\boldsymbol{R}}_i\rangle$ for $\hat{\boldsymbol{R}}_i = \hat{\boldsymbol{x}}_i$. Therefore, we conclude that

$$\text{Reg}_i^T(\hat{\boldsymbol{x}}_i) \leq \frac{1}{2\eta}\|\hat{\boldsymbol{z}}_i^0 - \hat{\boldsymbol{x}}_i\|_2^2.$$

This concludes the proof of Theorem 5.3. $\qquad\square$

# G Proof of Theorem 5.5

*Proof of Theorem 5.5.* At iteration $t \geq 1$, let $\boldsymbol{w}^t \in \mathcal{X}_{\geq}$ such that $\|\boldsymbol{w}^t - \Pi_{\boldsymbol{z}^{t-1}, \mathcal{X}_{\geq}}(\eta F(\boldsymbol{w}^t))\|_2 \leq \epsilon^{(t)}$. Then the first order optimality condition gives, for $\boldsymbol{z}^t = \Pi_{\boldsymbol{z}^{t-1}, \mathcal{X}_{\geq}}(\eta F(\boldsymbol{w}^t))$,

$$\langle \eta F(\boldsymbol{w}^t), \hat{\boldsymbol{z}} - \boldsymbol{z}^t\rangle \geq \frac{1}{2}\|\boldsymbol{z}^t - \hat{\boldsymbol{z}}\|_2^2 - \frac{1}{2}\|\boldsymbol{z}^{t-1} - \hat{\boldsymbol{z}}\|_2^2 + \frac{1}{2}\|\boldsymbol{z}^t - \boldsymbol{z}^{t-1}\|_2^2, \forall \hat{\boldsymbol{z}} \in \mathcal{X}_{\geq}.$$

Let us fix a player $i \in \{1, ..., n\}$. We apply the inequality above with $\hat{\boldsymbol{z}}_j = \boldsymbol{z}_j^t$ for $j \neq i$. This yields, for $\boldsymbol{x}_p = \boldsymbol{g}(\boldsymbol{w}_p^t), \forall p \in \{1, ..., n\}$ and $\boldsymbol{\ell}_i^t = -\nabla_{\boldsymbol{x}_i} u_i(\boldsymbol{x})$,

$$\langle \eta \boldsymbol{f}(\boldsymbol{g}(\boldsymbol{w}_i^t), \boldsymbol{\ell}_i^t), \hat{\boldsymbol{z}}_i - \boldsymbol{z}_i^t\rangle \geq \frac{1}{2}\|\boldsymbol{z}_i^t - \hat{\boldsymbol{z}}_i\|_2^2 - \frac{1}{2}\|\boldsymbol{z}_i^{t-1} - \hat{\boldsymbol{z}}_i\|_2^2 + \frac{1}{2}\|\boldsymbol{z}_i^t - \boldsymbol{z}_i^{t-1}\|_2^2, \forall \hat{\boldsymbol{z}}_i \in \Delta^{d_i}.$$

We now upper bound the left-hand side of the previous inequality. Note that

$$\langle \eta \boldsymbol{f}(\boldsymbol{g}(\boldsymbol{w}_i^t), \boldsymbol{\ell}_i^t), \hat{\boldsymbol{z}}_i - \boldsymbol{z}_i^t\rangle = \langle \eta \boldsymbol{f}(\boldsymbol{g}(\boldsymbol{w}_i^t), \boldsymbol{\ell}_i^t), \hat{\boldsymbol{z}}_i - \boldsymbol{w}_i^t\rangle + \langle \eta \boldsymbol{f}(\boldsymbol{g}(\boldsymbol{w}_i^t), \boldsymbol{\ell}_i^t), \boldsymbol{w}_i^t - \boldsymbol{z}_i^t\rangle.$$

Cauchy-Schwarz inequality ensures that

$$\langle \eta \boldsymbol{f}(\boldsymbol{g}(\boldsymbol{w}_i^t), \boldsymbol{\ell}_i^t), \boldsymbol{w}_i^t - \boldsymbol{z}_i^t\rangle \leq \eta\|\boldsymbol{f}(\boldsymbol{g}(\boldsymbol{w}_i^t), \boldsymbol{\ell}_i^t)\|_2\|\boldsymbol{w}_i^t - \boldsymbol{z}_i^t\|_2.$$

Note that $\Pi_{\boldsymbol{z}^{t-1}, \mathcal{X}_{\geq}}(\eta F(\boldsymbol{w}^t)) = \boldsymbol{z}^t$, so that $\|\boldsymbol{w}_i^t - \boldsymbol{z}_i^t\|_2 \leq \epsilon^{(t)}$. To bound $\|\boldsymbol{f}(\boldsymbol{g}(\boldsymbol{w}_i^t), \boldsymbol{\ell}_i^t)\|_2$, we note that by definition,

$$\|\boldsymbol{f}(\boldsymbol{x}_i, \boldsymbol{\ell}_i)\|_2^2 = \sum_{j=1}^{d_i}\left((\boldsymbol{x}_i - \boldsymbol{e}_j)^\top \boldsymbol{\ell}_i\right)^2 \leq \sum_{j=1}^{d_i}\|\boldsymbol{x}_i - \boldsymbol{e}_j\|_2^2\|\boldsymbol{\ell}_i\|_2^2 \leq 4d_i B_u^2.$$

This gives

$$\|\boldsymbol{f}(\boldsymbol{g}(\boldsymbol{w}_i^t), \boldsymbol{\ell}_i^t)\|_2 \leq 2B_u\sqrt{d_i}.$$

Overall, we have obtained that for all $\hat{\boldsymbol{z}}_i \in \Delta^{d_i}_{\geq}$, we have

$$\langle \eta \boldsymbol{f}(\boldsymbol{g}(\boldsymbol{w}_i^t), \boldsymbol{\ell}_i^t), \boldsymbol{w}_i^t - \hat{\boldsymbol{z}}_i\rangle \leq -\frac{1}{2}\|\boldsymbol{z}_i^t - \hat{\boldsymbol{z}}_i\|_2^2 + \frac{1}{2}\|\boldsymbol{z}_i^{t-1} - \hat{\boldsymbol{z}}_i\|_2^2 - \frac{1}{2}\|\boldsymbol{z}_i^t - \boldsymbol{z}_i^{t-1}\|_2^2 + \eta 2B_u\sqrt{d_i}\epsilon^{(t)}.$$

We sum the previous inequality for $t = 1, ..., T$ to obtain that for all $\hat{\boldsymbol{z}}_i \in \Delta^{d_i}_{\geq}$, we have

$$\sum_{t=1}^{T}\langle \eta \boldsymbol{f}(\boldsymbol{g}(\boldsymbol{w}_i^t), \boldsymbol{\ell}_i^t), \boldsymbol{w}_i^t - \hat{\boldsymbol{z}}_i\rangle \leq \frac{1}{2}\|\boldsymbol{z}_i^0 - \hat{\boldsymbol{z}}_i\|_2^2 - \frac{1}{2}\|\boldsymbol{z}_i^T - \hat{\boldsymbol{z}}_i\|_2^2 - \sum_{t=1}^{T}\frac{1}{2}\|\boldsymbol{z}_i^t - \boldsymbol{z}_i^{t-1}\|_2^2 + \eta 2B_u\sqrt{d_i}\sum_{t=1}^{T}\epsilon^{(t)}.$$

Overall, we conclude that

$$\sum_{t=1}^{T}\langle \boldsymbol{f}(\boldsymbol{g}(\boldsymbol{w}_i^t), \boldsymbol{\ell}_i^t), \boldsymbol{w}_i^t - \hat{\boldsymbol{z}}_i\rangle \leq \frac{1}{2\eta}\|\boldsymbol{z}_i^0 - \hat{\boldsymbol{z}}_i\|_2^2 + 2B_u\sqrt{d_i}\sum_{t=1}^{T}\epsilon^{(t)}, \forall \hat{\boldsymbol{z}}_i \in \Delta^{d_i}_{\geq}.$$

From Lemma 2.1 the left-hand side is equal to $\text{Reg}_i^T(\hat{\boldsymbol{x}}_i)$ for $\hat{\boldsymbol{x}}_i = \hat{\boldsymbol{z}}_i$. This concludes the proof of Theorem 5.5. $\qquad\square$

# H Proof of Theorem 5.6

*Proof of Theorem 5.5.* We will show that for any $\hat{\boldsymbol{w}} \in \mathcal{X}_{\geq}$, we have

$$\sum_{t=1}^{T} \left\langle F(\boldsymbol{w}^t), \boldsymbol{w}^t - \hat{\boldsymbol{w}} \right\rangle \leq \frac{1}{2\eta} \|\boldsymbol{w}^0 - \hat{\boldsymbol{w}}\|_2^2.$$

Since $\boldsymbol{x}_i^t = \boldsymbol{w}_i^t, \forall\, t \geq 1, \forall\, i \in \{1, ..., n\}$, this is enough to prove Theorem 5.5. Note that

$$\left\langle F(\boldsymbol{w}^t), \boldsymbol{w}^t - \hat{\boldsymbol{w}} \right\rangle = \left\langle F(\boldsymbol{w}^t), \boldsymbol{z}^t - \hat{\boldsymbol{w}} \right\rangle + \left\langle F(\boldsymbol{w}^t), \boldsymbol{w}^t - \boldsymbol{z}^t \right\rangle.$$

We will independently analyze each term of the right-hand side of the above equality.

For the first term, we note that the first-order optimality condition for $\boldsymbol{z}^t = \Pi_{\boldsymbol{z}^{t-1}, \mathcal{X}_{\geq}} (\eta F(\boldsymbol{w}^t))$ gives, for any $\hat{\boldsymbol{w}} \in \mathcal{X}_{\geq}$,

$$\left\langle \eta F(\boldsymbol{w}^t), \boldsymbol{z}^t - \hat{\boldsymbol{w}} \right\rangle \leq \frac{1}{2}\|\hat{\boldsymbol{w}} - \boldsymbol{z}^{t-1}\|_2^2 - \frac{1}{2}\|\hat{\boldsymbol{w}} - \boldsymbol{z}^t\|_2^2 - \frac{1}{2}\|\boldsymbol{z}^t - \boldsymbol{z}^{t-1}\|_2^2. \tag{12}$$

For the second term, we will prove the following lemma.

**Lemma H.1.** *Let $\eta > 0$ such that $\boldsymbol{w} \mapsto \eta F(\boldsymbol{w})$ is $1/\sqrt{2}$ Lipschitz continuous over $\mathcal{X}_{\geq}$. Then*

$$\left\langle \eta F(\boldsymbol{w}^t), \boldsymbol{w}^t - \boldsymbol{z}^t \right\rangle \leq \frac{1}{2}\|\boldsymbol{z}^t - \boldsymbol{z}^{t-1}\|_2^2. \tag{13}$$

*Proof of Lemma H.1.* We write

$$\left\langle \eta F(\boldsymbol{w}^t), \boldsymbol{w}^t - \boldsymbol{z}^t \right\rangle = \left\langle \eta F(\boldsymbol{z}^{t-1}), \boldsymbol{w}^t - \boldsymbol{z}^t \right\rangle + \left\langle \eta F(\boldsymbol{w}^t) - \eta F(\boldsymbol{z}^{t-1}), \boldsymbol{w}^t - \boldsymbol{z}^t \right\rangle.$$

We will bound independently each term in the above equation. From $\boldsymbol{w}^t = \Pi_{\boldsymbol{z}^{t-1}, \mathcal{X}_{\geq}} \left( \eta F(\boldsymbol{z}^{t-1}) \right)$ we have

$$\left\langle \eta F(\boldsymbol{z}^{t-1}), \boldsymbol{w}^t - \boldsymbol{z}^t \right\rangle \leq \frac{1}{2}\|\boldsymbol{z}^t - \boldsymbol{z}^{t-1}\|_2^2 - \frac{1}{2}\|\boldsymbol{z}^t - \boldsymbol{w}^t\|_2^2 - \frac{1}{2}\|\boldsymbol{w}^t - \boldsymbol{z}^{t-1}\|_2^2,$$

which gives

$$\left\langle \eta F(\boldsymbol{z}^{t-1}), \boldsymbol{w}^t - \boldsymbol{z}^t \right\rangle \leq \frac{1}{2}\|\boldsymbol{z}^t - \boldsymbol{z}^{t-1}\|_2^2 - \frac{1}{2}\|\boldsymbol{z}^t - \boldsymbol{w}^t\|_2^2 - \frac{1}{2}\|\boldsymbol{w}^t - \boldsymbol{z}^{t-1}\|_2^2, \tag{14}$$

From Cauchy-Schwarz inequality, we have

$$\left\langle \eta F(\boldsymbol{w}^t) - \eta F(\boldsymbol{z}^{t-1}), \boldsymbol{w}^t - \boldsymbol{z}^t \right\rangle \leq \|\eta F(\boldsymbol{w}^t) - \eta F(\boldsymbol{z}^{t-1})\|_2 \|\boldsymbol{w}^t - \boldsymbol{z}^t\|_2.$$

Recall that

$$\boldsymbol{w}^t = \Pi_{\boldsymbol{z}^{t-1}, \mathcal{X}} \left( \eta F(\boldsymbol{z}^{t-1}) \right)$$
$$\boldsymbol{z}^t = \Pi_{\boldsymbol{z}^{t-1}, \mathcal{X}} \left( \eta F(\boldsymbol{w}^t) \right)$$

Since the proximal operator is 1-Lipschitz continuous, and since $\boldsymbol{w} \mapsto \eta F(\boldsymbol{w})$ is $1/\sqrt{2}$-Lipschitz continuous, we obtain

$$\left\langle \eta F(\boldsymbol{w}^t) - \eta F(\boldsymbol{z}^{t-1}), \boldsymbol{w}^t - \boldsymbol{z}^t \right\rangle \leq \frac{1}{2}\|\boldsymbol{w}^t - \boldsymbol{z}^{t-1}\|_2^2. \tag{15}$$

We can now sum (14) and (15) to obtain

$$\left\langle \eta F(\boldsymbol{w}^t), \boldsymbol{w}^t - \boldsymbol{z}^t \right\rangle \leq \frac{1}{2}\|\boldsymbol{z}^t - \boldsymbol{z}^{t-1}\|_2^2 - \frac{1}{2}\|\boldsymbol{z}^t - \boldsymbol{w}^t\|_2^2 \leq \frac{1}{2}\|\boldsymbol{z}^t - \boldsymbol{z}^{t-1}\|_2^2.$$

$\square$

We have shown in Lemma 5.2 that $F$ is $L_F$-Lipschitz continuous for normal-form games. Our choice of step size $\eta = \frac{1}{L_F \sqrt{2}}$ ensures that that $\boldsymbol{\omega} \mapsto \eta F(\boldsymbol{w})$ is $1/\sqrt{2}$-Lipschitz continuous as in the assumptions of Lemma H.1.

Combining (12) with (13) yields

$$\langle \eta F(\boldsymbol{w}^t), \boldsymbol{w}^t - \hat{\boldsymbol{w}} \rangle \le \frac{1}{2}\|\hat{\boldsymbol{w}} - \boldsymbol{z}^{t-1}\|_2^2 - \frac{1}{2}\|\hat{\boldsymbol{w}} - \boldsymbol{z}^t\|_2^2.$$

Summing this inequality for $t = 1, ..., T$ and telescoping, we obtain

$$\sum_{t=1}^{T} \langle \eta F(\boldsymbol{w}^t), \boldsymbol{w}^t - \hat{\boldsymbol{w}} \rangle \le \frac{1}{2}\|\hat{\boldsymbol{w}} - \boldsymbol{z}^0\|_2^2 - \frac{1}{2}\|\hat{\boldsymbol{w}} - \boldsymbol{z}^T\|_2^2$$

which directly yields

$$\sum_{t=1}^{T} \langle \eta F(\boldsymbol{w}^t), \boldsymbol{w}^t - \hat{\boldsymbol{w}} \rangle \le \frac{1}{2}\|\hat{\boldsymbol{w}} - \boldsymbol{z}^0\|_2^2. \tag{16}$$

Overall, for any $\left(\hat{\boldsymbol{R}}_1, ..., \hat{\boldsymbol{R}}_n\right) \in \mathcal{X}_\ge$ we obtain that $\sum_{i=1}^{T} \mathrm{Reg}_i^T(\hat{\boldsymbol{R}}_i)$ is upper bounded by $\sum_{i=1}^{n} \frac{1}{2\eta}\|\boldsymbol{w}_i^0 - \hat{\boldsymbol{R}}_i\|_2^2$. Now from Lemma 2.1, for any $(\hat{\boldsymbol{x}}_1, ..., \hat{\boldsymbol{x}}_n) \in \Delta$, we conclude that

$$\sum_{i=1}^{T} \mathrm{Reg}_i^T(\hat{\boldsymbol{x}}_i) \le \sum_{i=1}^{n} \frac{1}{2\eta}\|\boldsymbol{w}_i^0 - \hat{\boldsymbol{x}}_i\|_2^2.$$

This concludes the proof of Theorem 5.6. $\qquad\qquad\square$

# I  Proof of Lemma 5.7

*Proof of Lemma 5.7.* The proof of Lemma 5.7 follows the lines of the proof of Lemma 5.2. Clearly, for matrix games we have $F = h \circ g$ with $h : \mathbb{R}^{d_1} \times \mathbb{R}^{d_2} \to \mathbb{R}^{d_1} \times \mathbb{R}^{d_2}$ defined as Proposition 1.

$$h \begin{bmatrix} \boldsymbol{x} \\ \boldsymbol{y} \end{bmatrix} = \begin{bmatrix} \boldsymbol{f}(\boldsymbol{x}, \boldsymbol{A}\boldsymbol{y}) \\ \boldsymbol{f}(\boldsymbol{y}, -\boldsymbol{A}^\top \boldsymbol{x}) \end{bmatrix} \tag{17}$$

The function $g$ is Lipschitz continuous over $\Delta_\ge^{d_1}$ (Proposition 1), with a Lipschitz constant of $L_g = \sqrt{d_1}$. Let us now compute the Lipschitz constant of $h$. Observe that:

$$\|\boldsymbol{f}(\boldsymbol{x}, \boldsymbol{A}\boldsymbol{y}) - \boldsymbol{f}(\boldsymbol{x}', \boldsymbol{A}\boldsymbol{y}')\|_2^2$$

$$= \sum_{i=1}^{d_1} \left((\boldsymbol{x} - \boldsymbol{e}_i)^\top \boldsymbol{A}\boldsymbol{y} - (\boldsymbol{x}' - \boldsymbol{e}_i)^\top \boldsymbol{A}\boldsymbol{y}'\right)^2$$

$$= \sum_{i=1}^{d_1} ((\boldsymbol{x} - \boldsymbol{e}_i)^\top \boldsymbol{A}\boldsymbol{y} - (\boldsymbol{x}' - \boldsymbol{e}_i)^\top \boldsymbol{A}\boldsymbol{y} + (\boldsymbol{x}' - \boldsymbol{e}_i)^\top \boldsymbol{A}\boldsymbol{y} - (\boldsymbol{x}' - \boldsymbol{e}_i)^\top \boldsymbol{A}\boldsymbol{y}')^2$$

$$= \sum_{i=1}^{d_1} \left((\boldsymbol{x} - \boldsymbol{x}')^\top \boldsymbol{A}\boldsymbol{y} + (\boldsymbol{x}' - \boldsymbol{e}_i)^\top \boldsymbol{A}(\boldsymbol{y} - \boldsymbol{y}')\right)^2$$

$$\le 2d_1 \left((\boldsymbol{x} - \boldsymbol{x}')^\top \boldsymbol{A}\boldsymbol{y}\right)^2 + 2 \sum_{i=1}^{d_1} \left((\boldsymbol{x}' - \boldsymbol{e}_i)^\top \boldsymbol{A}(\boldsymbol{y} - \boldsymbol{y}')\right)^2$$

$$\le 2d_1 \|\boldsymbol{A}\|_{op}^2 \|\boldsymbol{x} - \boldsymbol{x}'\|_2^2 + 4d_1 \|\boldsymbol{A}\|_{op}^2 \|\boldsymbol{y} - \boldsymbol{y}'\|_2^2.$$

Similarly, we have that $\|\boldsymbol{f}(\boldsymbol{y}, -\boldsymbol{A}^\top \boldsymbol{x}) - \boldsymbol{f}(\boldsymbol{y}', -\boldsymbol{A}^\top \boldsymbol{x}')\|_2^2$ is upper bounded by

$$2d_2 \|\boldsymbol{A}\|_{op}^2 \|\boldsymbol{y} - \boldsymbol{y}'\|_2^2 + 4d_2 \|\boldsymbol{A}\|_{op}^2 \|\boldsymbol{x} - \boldsymbol{x}'\|_2^2,$$

and thus

$$\left\| h \begin{bmatrix} \boldsymbol{x} \\ \boldsymbol{y} \end{bmatrix} - h \begin{bmatrix} \boldsymbol{x}' \\ \boldsymbol{y}' \end{bmatrix} \right\|_2 \le \|\boldsymbol{A}\|_{op} \sqrt{6 \max\{d_1, d_2\}} \left\| \begin{bmatrix} \boldsymbol{x} \\ \boldsymbol{y} \end{bmatrix} - \begin{bmatrix} \boldsymbol{x}' \\ \boldsymbol{y}' \end{bmatrix} \right\|_2.$$

Therefore, the Lipschitz constant of $h$ is $L_h = \|\boldsymbol{A}\|_{op} \sqrt{6 \max\{d_1, d_2\}}$.

Since $F = h \circ g$, we obtain that the Lipschitz constant $L_F$ of $F$ is $L_F = L_h \times L_g = \sqrt{6}\|\boldsymbol{A}\|_{op} \max\{d_1, d_2\}$. $\qquad\square$

## J   Extensive-Form Games

In this section we show how to extend our convergence results for Conceptual RM$^+$ from normal-form games to EFGs. Briefly, an EFG is a game played on a tree, where each node belongs to some player, and the player chooses a probability distribution over branches. Moreover, players have *information sets*, which are groups of nodes belonging to a player such that they cannot distinguish among them, and thus they must choose the same probability distribution at all nodes in an information set. When a leaf $h$ is reached, each player $i$ receives some payoff $v_i(h) \in [0,1]$. In order to extend our results, we will use the CFR regret decomposition [37, 9], and then show how to run the Conceptual RM$^+$ algorithm on the resulting set of strategy spaces (which will be a Cartesian product of positive orthants). The CFR regret decomposition works in the space of *behavioral strategies*, which represents the strategy space of each player as a Cartesian product of simplices, with each simplex corresponding to the set of possible ways to randomize over actions at a given information set for the player. Formally, we write the polytope of behavioral-form strategies as

$$\mathcal{X} = \times_{i \in [n], j \in \mathcal{D}_i} \Delta^{n_j},$$

where $\mathcal{D}_i$ is the set of information sets for player $i$ and $n_j$ is the number of actions at information set $j$. Let $P = \sum_{i \in [n], j \in \mathcal{D}_i} n_j$ be the dimension of $\mathcal{X}$. In EFGs with *perfect recall*, meaning that a player never forgets something they knew in the past, the *sequence-form* is an equivalent representation of the set of strategies, which allows one to write the payoffs for each player as a multilinear function. This in turn enables optimization and regret minimization approaches that exploit multilinearity, e.g. bilinearity in the two-player zero-sum setting [21, 25, 10]. Instead of working on this representation, the CFR approach minimizes a notion of local regret at each information set, using so-called *counterfactual values*. The weighted sum of counterfactual regrets at each information set is an upper bound on the sequence-form regret [37], and thus a player in an EFG can minimize their regret by locally minimizing each counterfactual regret. Informally, the counterfactual value is the expected value of an action at a information set, conditional on the player at the information set playing to reach that information set and then taking the corresponding action. The counterfactual value associated to each tuples of player $i$, information set $j \in \mathcal{D}_i$, and action $a \in A_j$ is $G_{ija}(\boldsymbol{x}) := \sum_{h \in \mathcal{L}_{ja}} \prod_{(\hat{j}, \hat{a}) \in \mathcal{P}_j(h)} \boldsymbol{x}[\hat{j}, \hat{a}] v_i(h)$, where $\mathcal{L}_{ja}$ is the set of leaf nodes reachable from information set $j$ after taking action $a$, and $\mathcal{P}_j(h)$ is the set of pairs of information sets and actions $(\hat{j}, \hat{a})$ on the path from the root to $h$, except that information sets belonging to player $i$ are excluded, unless they occur *after* $j, a$.

We will be concerned with the counterfactual regret, given by the operator $H : \mathcal{X} \to \mathbb{R}^{\sum_{i \in [n], j \in \mathcal{D}_i} n_j}$ defined as $H_{ija}(\boldsymbol{x}) := G_{ija}(\boldsymbol{x}) - \langle G_{ij}(\boldsymbol{x}), \boldsymbol{x}^j \rangle$. Now we can show that the counterfactual regret operator $H$ is Lipschitz continuous. Intuitively, this should hold since $H$ is multilinear.

**Lemma J.1.** *For any behavioral strategies $\boldsymbol{x}, \boldsymbol{x}' \in \mathcal{X}$, $\|H(\boldsymbol{x}) - H(\boldsymbol{x}')\|_2 \leq \sqrt{2P}\|\boldsymbol{x} - \boldsymbol{x}'\|_2$.*

*Proof.* We start by showing a bound for $G$. We first analyze the change in a single coordinate of $G$ for a given $i \in [n], j \in \mathcal{D}_i, a \in A_j$. We focus on how $G_{ija}$ changes with respect to the change in $|\boldsymbol{x}[\hat{j}, \hat{a}] - \boldsymbol{x}'[\hat{j}, \hat{a}]|$ for some arbitrary information set-action pair $(\hat{j}, \hat{a}) \in \mathcal{P}_j(h)$ for some $h \in \mathcal{L}_{ja}$. To alleviate inline notation, let $\mathcal{P}_j^{\hat{j}, \hat{a}}(h) = \mathcal{P}_j(h) \setminus \{(\hat{j}, \hat{a})\}$.

$$
\begin{aligned}
G_{ija}(x) = &\ \boldsymbol{x}[\hat{j}, \hat{a}] \sum_{h \in \mathcal{L}_{\hat{j}\hat{a}} \cap \mathcal{L}_{ja}} \prod_{(\bar{j}, \bar{a}) \in \mathcal{P}_j^{\hat{j}, \hat{a}}(h)} \boldsymbol{x}[\bar{j}, \bar{a}] v_i(h) \\
&+ \sum_{h \in \mathcal{L}_{ja} \setminus \mathcal{L}_{\hat{j}\hat{a}}} \prod_{(\bar{j}, \bar{a}) \in \mathcal{P}_j(h)} \boldsymbol{z}[\bar{j}, \bar{a}] v_i(h) \\
\leq &\ |\boldsymbol{x}[\hat{j}, \hat{a}] - \boldsymbol{x}'[\hat{j}, \hat{a}]| \sum_{h \in \mathcal{L}_{\hat{j}\hat{a}} \cap \mathcal{L}_{ja}} \prod_{(\bar{j}, \bar{a}) \in \mathcal{P}_j^{\hat{j}, \hat{a}}(h)} \boldsymbol{x}[\bar{j}, \bar{a}] v_i(h) \\
&+ \boldsymbol{x}'[\hat{j}, \hat{a}] \sum_{h \in \mathcal{L}_{\hat{j}\hat{a}} \cap \mathcal{L}_{ja}} \prod_{(\bar{j}, \bar{a}) \in \mathcal{P}_j(h)} \boldsymbol{x}[\bar{j}, \bar{a}] v_i(h) \\
&+ \sum_{h \in \mathcal{L}_{ja} \setminus \mathcal{L}_{\hat{j}\hat{a}}} \prod_{(\bar{j}, \bar{a}) \in \mathcal{P}_j(h)} \boldsymbol{x}[\bar{j}, \bar{a}] v_i(h)
\end{aligned}
$$

Now let us bound the error term by noting that $v_i(h) \leq 1$ for all $h$ by assumption:

$$|\boldsymbol{x}[\hat{j}, \hat{a}] - \boldsymbol{x}'[\hat{j}, \hat{a}]| \sum_{h \in \mathcal{L}_{\hat{j}\hat{a}} \cap \mathcal{L}_{ja}} \prod_{(\bar{j}, \bar{a}) \in \mathcal{P}_j^{\hat{j}, \hat{a}}(h)} \boldsymbol{x}[\bar{j}, \bar{a}] v_i(h)$$

$$\leq |\boldsymbol{x}[\hat{j}, \hat{a}] - \boldsymbol{x}'[\hat{j}, \hat{a}]| \sum_{h \in \mathcal{L}_{\hat{j}\hat{a}} \cap \mathcal{L}_{ja}} \prod_{(\bar{j}, \bar{a}) \in \mathcal{P}_j^{\hat{j}, \hat{a}}(h)} \boldsymbol{x}[\bar{j}, \bar{a}]$$

$$\leq |\boldsymbol{x}[\hat{j}, \hat{a}] - \boldsymbol{x}'[\hat{j}, \hat{a}]|,$$

where the last inequality is because the sum of reach probabilities on leaf nodes in $\mathcal{L}_{\hat{j}\hat{a}} \cap \mathcal{L}_{ja}$ after conditioning on player $i$ playing to reach $(j, a)$ and $(\hat{j}, \hat{a})$ being played with probability one, is less than or equal to one.

By iteratively applying this argument to each $(\hat{j}, \hat{a}) \in \mathcal{P}_j(h)$, we get

$$G_{ija}(\boldsymbol{x}) \leq G_{ija}(\boldsymbol{x}') + \sum_{h \in \mathcal{L}_{ja}} \sum_{(\hat{j}, \hat{a}) \in \mathcal{P}_j(h)} |\boldsymbol{x}[\hat{j}, \hat{a}] - \boldsymbol{x}'[\hat{j}, \hat{a}]| \qquad (18)$$

$$\leq G_{ija}(\boldsymbol{x}') + \|\boldsymbol{x} - \boldsymbol{x}'\|_1.$$

Repeating the same argument for $\boldsymbol{x}'$ gives

$$|G_{ija}(\boldsymbol{x}) - G_{ija}(\boldsymbol{x}')| \leq \|\boldsymbol{x} - \boldsymbol{x}'\|_1.$$

Secondly, we bound the difference in the inner product terms.

$$\langle G_{ij}(\boldsymbol{x}), \boldsymbol{x}^j \rangle = \sum_{a \in A_j} \boldsymbol{x}[j, a] G_{ija}(\boldsymbol{x})$$

$$\leq \sum_{a \in A_j} \left[ |\boldsymbol{x}[j, a] - \boldsymbol{x}'[j, a]| + \boldsymbol{x}'[j, a] G_{ija}(\boldsymbol{x}) \right]$$

$$\leq \|\boldsymbol{x}^j - \boldsymbol{x}^{j'}\|_1 + \sum_{a \in A_j} \boldsymbol{x}'[j, a] G_{ija}(\boldsymbol{x}') + \sum_{a \in A_j} \boldsymbol{x}'[j, a] \sum_{h \in \mathcal{L}_{ja}} \sum_{(\hat{j}, \hat{a}) \in \mathcal{P}_j(h)} |\boldsymbol{x}[\hat{j}, \hat{a}] - \boldsymbol{x}'[\hat{j}, \hat{a}]|$$

$$\leq \langle G_{ij}(\boldsymbol{x}'), \boldsymbol{x}' \rangle + \|\boldsymbol{x} - \boldsymbol{x}'\|_1$$

where the second-to-last line is by Eq. (18). Again we can start from $\boldsymbol{x}'$ instead to get

$$|\langle G_{ij}(\boldsymbol{x}), \boldsymbol{x}^j \rangle - \langle G_{ij}(\boldsymbol{x}'), \boldsymbol{x}' \rangle| \leq \|\boldsymbol{x} - \boldsymbol{x}'\|_1.$$

Putting together all our bonds and applying norm equivalence, we get that

$$\|H(\boldsymbol{x}) - H(\boldsymbol{x}')\|_2^2 \leq \sum_{i \in [n]} \sum_{j \in \mathcal{D}_i, a \in A_j} 2\|\boldsymbol{x} - \boldsymbol{x}'\|_1^2$$

$$\leq 2P\|\boldsymbol{x} - \boldsymbol{x}'\|_2^2.$$

Taking square roots completes the proof. $\qquad \square$

Since we want to run smooth RM$^+$, we will need to consider the lifted strategy space for each decision point. Let $\mathcal{Z}$ be the Cartesian product of the positive orthants for each information set, i.e. $\mathcal{Z} = \times_{i \in [n], j \in \mathcal{D}_i} \mathbb{R}_+^{n_j}$. Now let $\hat{g} : \mathcal{Z} \to \mathcal{X}$ be the function that normalizes each vector from the positive orthant to the simplex such that we get a behavioral strategy, i.e. $\hat{g}_j(\boldsymbol{z}) = g(\boldsymbol{z}^j)$, where $\boldsymbol{z}^j$ is the slice of $\boldsymbol{z}$ corresponding to information set $j$. The function $\hat{g}$ is also Lipschitz continuous.

**Lemma J.2.** *Suppose that $\boldsymbol{z}, \boldsymbol{z}' \in \mathcal{Z}$ satisfy $\|\boldsymbol{z}^j\|_1 \geq R_{0,j}, \|\boldsymbol{z}^{j'}\|_1 \geq R_{0,j}$ for all $i \in [n], j \in \mathcal{D}_i$. Then, $\|\hat{g}(\boldsymbol{z}) - \hat{g}(\boldsymbol{z}')\|_2 \leq \max_{i \in [n], j \in \mathcal{D}_i} \sqrt{n_j / R_{0,j}} \|\boldsymbol{z} - \boldsymbol{z}'\|_2$.*

*Proof.* We have from Proposition 1 that

$$\|\hat{\boldsymbol{g}}(\boldsymbol{z}) - \hat{\boldsymbol{g}}(\boldsymbol{z}')\|_2^2 = \sum_{i \in [n], j \in \mathcal{D}_i} \|\boldsymbol{g}(\boldsymbol{z}) - \boldsymbol{g}(\boldsymbol{z}')\|_2^2$$

$$\leq \sum_{i \in [n], j \in \mathcal{D}_i} n_j / R_{0,j} \|\boldsymbol{z}^j - \boldsymbol{z}^{j'}\|_2^2$$

$$\leq \max_{i \in [n], j \in \mathcal{D}_i} n_j / R_{0,j} \|\boldsymbol{z} - \boldsymbol{z}'\|_2^2$$

$\square$

Now let us introduce the operator $F : \mathcal{Z} \to \mathbb{R}^{\sum_{i \in [n], j \in \mathcal{D}_i} n_j}$ for EFGs. For a given $\boldsymbol{z} \in \mathcal{Z}$, the operator will output the regret associated with the counterfactual values for each decision set $j$. $F$ will be composed of two functions, first $\hat{g}$ maps a given $\boldsymbol{z}$ to some behavioral strategy $\boldsymbol{x} = \hat{g}(\boldsymbol{z})$, and then the operator $H : \mathcal{X} \to \mathbb{R}^{\sum_{i \in [n], j \in \mathcal{D}_i} n_j}$ outputs the regrets for the counterfactual values.

Now we can apply our bounds on the Lipschitz constant for $\hat{g}$ and $H$ to get that $F$ is Lipschitz continuous with Lipschitz constant $2P \max_{i \in [n], j \in \mathcal{D}_i} \sqrt{n_j / R_{0,j}}$. Combining our Lipschitz result with our setup of $\mathcal{X}$ and $F$, we can now run Algorithm 4 on $\mathcal{X}$ and $F$ and apply Theorem 5.5 to get a smooth-RM$^+$-based algorithm that allows us to compute a sequence of iterates with regret at most $\epsilon$ in at $O(1/\epsilon)$ iterations and using $O(\log(1/\epsilon)/\epsilon)$ gradient computations.

# K   Details on the Numerical Experiments

**Efficient orthogonal projection on $\Delta_{\geq}^n$.**   Recall that $\Delta_{\geq}^n = \{\boldsymbol{R} \in \mathbb{R}^n \mid \boldsymbol{R} \geq \boldsymbol{0}, \boldsymbol{1}_n^\top \boldsymbol{R} \geq 1\}$. Let $\boldsymbol{y} \in \mathbb{R}^n$ and let us consider

$$\min_{\boldsymbol{x} \geq \boldsymbol{0}, \boldsymbol{1}_n^\top \boldsymbol{x} \geq 1} \frac{1}{2} \|\boldsymbol{x} - \boldsymbol{y}\|_2^2.$$

Introducing a Lagrange multiplier $\mu \geq 0$ for the constraint $1 - \boldsymbol{1}_n^\top \boldsymbol{x} \leq 0$, we arrive at

$$\min_{\boldsymbol{x} \geq \boldsymbol{0}} \max_{\mu \geq 0} \frac{1}{2} \|\boldsymbol{x} - \boldsymbol{y}\|_2^2 + \mu \left(1 - \boldsymbol{1}_n^\top \boldsymbol{x}\right).$$

Let us call $(\boldsymbol{x}, \mu) \in \mathbb{R}_+^n \times \mathbb{R}_+$ an optimal solution to the above saddle-point problem. Stationarity of the Lagrangian function shows that $x_i = [y_i + \mu]^+, \forall\, i \in [n]$. Therefore, we could simply use binary search to solve the following univariate concave problem:

$$\max_{\mu \geq 0} \mu - \frac{1}{2} \|[\boldsymbol{y} + \mu \boldsymbol{1}_n]^+\|_2^2.$$

Let us use the Karush-Kuhn-Tucker conditions. Complementary slackness gives $\mu \cdot \left(1 - \boldsymbol{1}_n^\top \boldsymbol{x}\right) = 0$. If $\mu = 0$, then $\boldsymbol{x} = [\boldsymbol{y}]^+$, and by primal feasibility we must have $\boldsymbol{1}_n^\top \boldsymbol{x} \geq 1$, i.e., $\boldsymbol{1}_n^\top [\boldsymbol{y}]^+ \geq 1$. If that is not the case, then we can not have $\mu = 0$, and we must have $1 - \boldsymbol{1}_n^\top \boldsymbol{x} = 0$, i.e., $\boldsymbol{x} \in \Delta^n$. In this case, we obtain that $\boldsymbol{x}$ is the orthogonal projection of $\boldsymbol{y}$ on $\Delta^n$. Overall, we see that $\boldsymbol{x}$ is always either $[\boldsymbol{y}]^+$, the orthogonal projection of $\boldsymbol{y}$ on $\mathbb{R}_+^n$, or $\boldsymbol{x}$ is the orthogonal projection of $\boldsymbol{y}$ on $\Delta^n$. Since $\Delta_{\geq}^n \subset \mathbb{R}_+^n$, we can compute the orthogonal projection on $\Delta_{\geq}^n$ as follows:

Compute $\boldsymbol{x} = [\boldsymbol{y}]^+$. If $\boldsymbol{1}_n^\top \boldsymbol{x} \geq 1$, then we have found the orthogonal projection of $\boldsymbol{y}$ on $\Delta_{\geq}^n$. Else, return the orthogonal projection of $\boldsymbol{y}$ on the simplex $\Delta^n$.

## K.1   Performances of ExRM$^+$, Stable PRM$^+$ and Smooth PRM$^+$ on our small matrix game example

In this section we provide detailed numerical results for ExRM$^+$, Stable PRM$^+$, and Smooth PRM$^+$ on our $3 \times 3$ matrix-game counterexample. All algorithms use linear averaging and Stable and Smooth PRM$^+$ use alternation. We choose a step size of $\eta = 0.1$ for our implementation of these algorithms. The results are presented in Figure 6 for ExRM$^+$, in Figure 7 for Stable PRM$^+$ and in Figure 8 for Smooth PRM$^+$.

## K.2   Extensive-form game used in the experiments

We used the following games in the experiments:

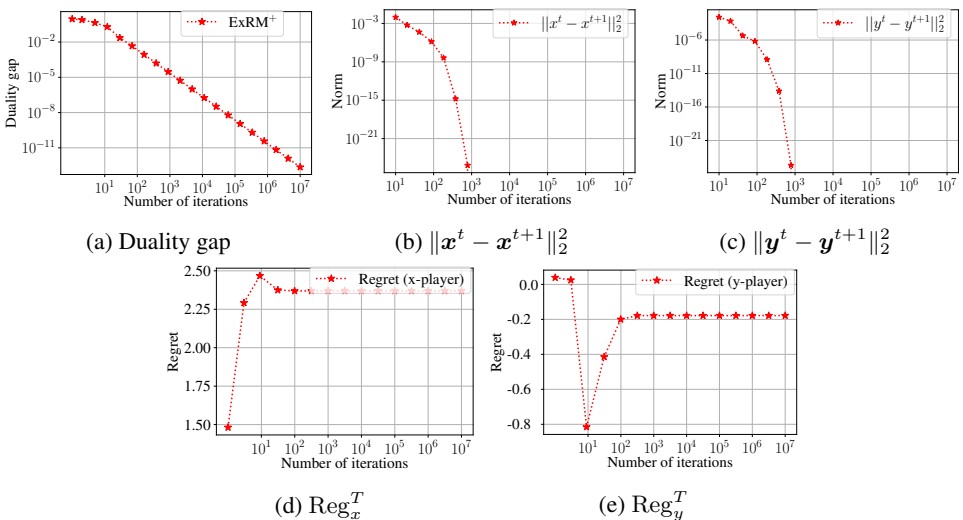

Figure 6: Empirical performance of ExRM$^+$ (with linear averaging) on our $3 \times 3$ matrix game from Section 2.

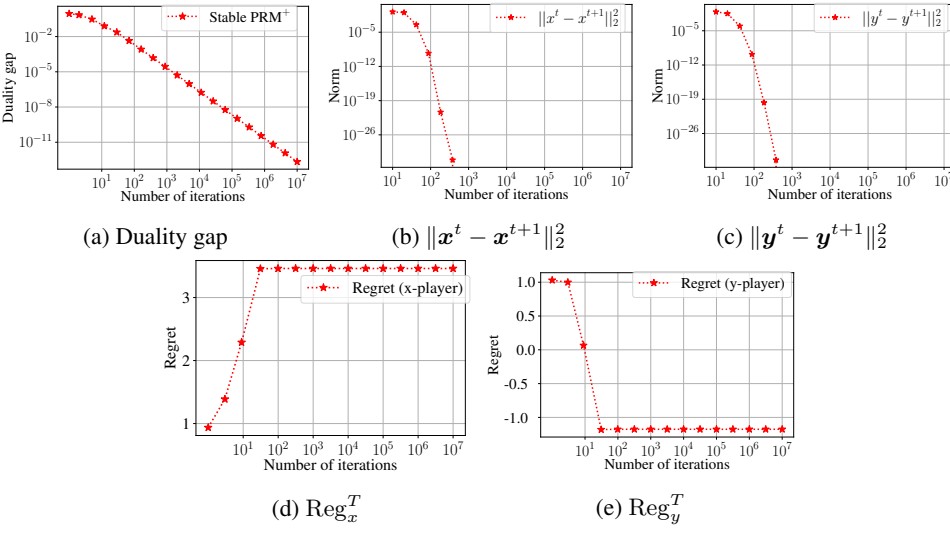

Figure 7: Empirical performance of Stable PRM$^+$ (with alternation and linear averaging) on our $3 \times 3$ matrix game from Section 2.

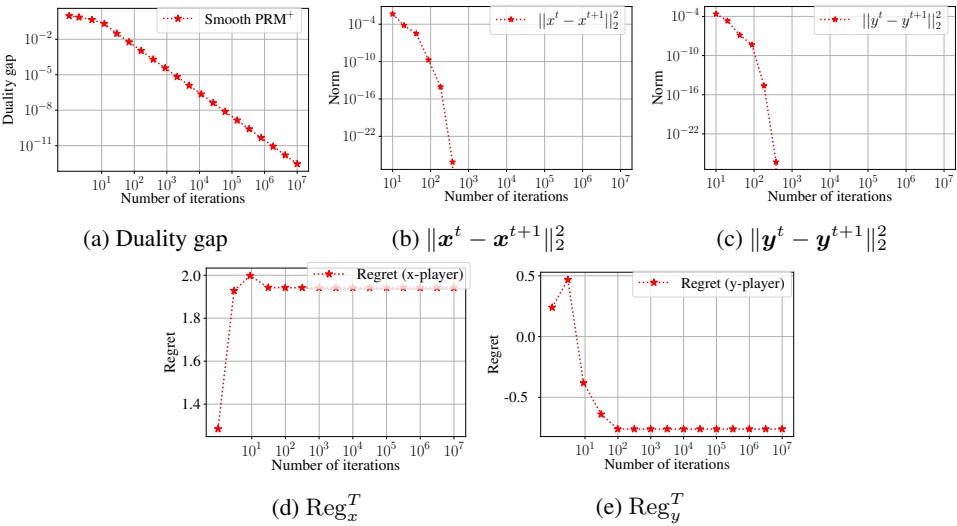

Figure 8: Empirical performance of Smooth PRM$^+$ (with alternation and linear averaging) on our $3 \times 3$ matrix game from Section 2.

- 2-player Sheriff is a two-player general-sum game inspired by the Sheriff of Nottingham board game. It was introduced as a benchmark for correlated equilibria by Farina et al. [11]. The variant of the game we use has the following parameters:
  - maximum number of items that can be smuggled: 10
  - maximum bribe amount: 3
  - number of bargaining rounds: 3
  - value of each item: 5
  - penalty for illegal item found in cargo: 1
  - penalty for Sheriff if no illegal item found in cargo: 1

  The number of nodes in this game is 9648.
- 3-player Leduc poker is a 3-player version of the standard benchmark of Leduc poker [33]. The game has 15659 nodes.
- 4-player Kuhn poker is a 4-player version of the standard benchmark of Kuhn poker [26]. We use a larger variant the standard one, to assess the scalability of our algorithm. The variant we use has six ranks in the deck. The game has 23402 nodes.
- 4-player Liar's dice is a 4-player version of the game Liar's dice, already used as a benchmark by Lisỳ et al. [27]. We use a variant with 1 die per player, each with two distinct faces. The game has 8178 nodes.

