# OpenReview forum: "Regret Matching+: (In)Stability and Fast Convergence in Games"
_NeurIPS.cc/2023/Conference — NeurIPS 2023 spotlight_

### Official Review · Reviewer_QXqy · 2023-07-01

**Soundness:** 3 good
**Presentation:** 3 good
**Contribution:** 4 excellent
**Rating:** 7
**Confidence:** 4

**Summary:**

The paper pushed RM+ based algorithm better than $O(\sqrt T)$ theoretically the first time. Specifically, the authors proposed smooth RM+, conceptual RM+ and extragradient RM+ with theoretical regret guarantee.

**Strengths:**

To the best of my knowledge, this paper is the first one to establish RM+ based algorithm with better theoretical gurantee than $O(\sqrt T)$. Before, although people observe $O(1)$ regret guarantee of RM+ in *most* games (there exist counterexamples that make it around $O(T^{0.7})$), there's no variant of RM+ that has better than $O(\sqrt T)$ regret bound.

However, this paper proposed a really simple and straight-forward version of RM+ and proved better regret bound. This may partially reveal why RM+ and CFR+ have good performance in many games.

**Weaknesses:**

- The main text of the paper does not reveal much about the proof intuition.

**Questions:**

I'm a bit confused about the CFR part. Does CFR equipped with the new RM+ has constant social regret?

**Limitations:**

The authors discussed limitations in the conclusion.

---

> ### Author Rebuttal · Authors · 2023-08-09
>
> We thank the reviewer for the valuable feedback. Please see below for our response.
>
> - _The proof intuition in the main text:_ We will make sure to provide more intuition of the proofs in the final version when an additional page is available.
> - _Does CFR equipped with the new RM+ have constant social regret?_ Yes, as we mentioned at the end of Section 5, the Clairvoyant CFR algorithm that computes a sequence of iterates with regret at most $\epsilon$ in $O(1/\epsilon)$ iterations using $O(\log(1/\epsilon)/\epsilon)$ gradient computations. Proofs for extensive-form games are in Appendix J.

---

> > ### Comment · Reviewer_QXqy · 2023-08-10
> > **Re: Rebuttal by Authors**
> >
> > Thank you for answering my questions !

---

### Official Review · Reviewer_GXrh · 2023-07-07

**Soundness:** 3 good
**Presentation:** 3 good
**Contribution:** 3 good
**Rating:** 7
**Confidence:** 3

**Summary:**

The authors study $\text{RM}^+$. They show that there are instances of loss sequences in variants of $\text{RM}^+$ unstable. The authors point out that the decisions of $\text{RM}^+$-based algorithms are performed by normalizing an aggregate payoff vector. Hence, if one inputs an instance that is "close" to the origin, two consecutive aggregate payoffs can point in different directions in spite of being relatively close, which could make the algorithm cycle between two strategies. The authors exhibit an example establishing that the potential instability mentioned does indeed occur.

The authors then present two methods to the instability discussed dealing with having two consecutive aggregate payoffs being too close to the origin. In one, they simply re-initialize the regret vector too some non-zero amount. The other solution is simply to ignore an area of the space near the origin and projecting onto that space. Not surprisingly, the second method seems to have a better theoretical guarantee but it's more expensive to implement.

**Strengths:**

The paper is very-well written. It was a pleasure to read. I had some very minor comments about the draft in regards to exposition and writing.

I also like how natural their solutions are to the issue of instability. The proofs seem to be correct and everything feels very natural, which is probably due to the authors' writing. The experiments also seem reasonable and establish that their methods perform well with random synthetic data.

**Weaknesses:**

Would the authors please put their contribution in context? How "important" it is to the overall community? They have already addressed some of this in the draft, but I was hoping for longer explanation in the rebuttal period.

Some small suggestions
1. Generally, I think citations are not appropriate for abstracts since abstracts are often read independently of the full paper, so readers may not have access to the cited sources at that stage. If you want to include a specific reference or highlight previous related work in the abstract, I recommend paraphrasing the information or briefly mention the authors' names and year of publication without the specific citation. This maintains the abstract's clarity and conciseness while still giving credit to the relevant work. I realize adding the references simply as numbers might have done intentionally to hide the identity of the authors.
2. I'd recommend combining the sentences for on lines 45 to 48 "However, in a game setting..." and "Indeed, we identify..." to avoid using the word "this" which is a bit ambiguous.
3. Would it be possible to list the references in increasing order?
4. For the sake of completeness, it would be useful if T were defined as the horizon or something like that before being used.
5. Notation-wide, I think $\Delta(3)$ should probably be $\Delta^3$ on line 169.
6. The definition of $\mathbf{z}^t$ in Theorem 4.1 was a bit unclear to me until I saw it defined in Algorithm, 1. Maybe declare it before like you did for Theorem 5.1 in expression (2)
7. On line 346, when you say "$\text{RM}^+$-based algorithms" I would add in parenthesis to refer to which algorithms are meant (either by number or name)

**Questions:**

Q1: More the smooth predictive RM^+, was there anything essentially different if one uses a different threshold for $||R_1||$? Now, 1 is being used, which seems very reasonable. However, how ``far away" does one need to know from 0 to help stabilization? Like if I were to change 1 for a constant C, how does this affect the theoretical guarantees?

Q2: I must be missing something, but I don't fully understand why clairvoyant CFR doesn't seem to perform as well with EFG as it does with matrix games. If you were to use the strategic game equivalent of the EFGs tested, what is the performance. I apologize because there is probably something I'm missing.

Q3: In your experiments, it seems the data was generated randomly. This is good to know. However, did you try to create data that would be more "difficult" like maybe a space that is an annulus?

**Limitations:**

The authors make an effort of what they have consider and the limitations of their work in terms of experimental results obtained for EFGs

I also would like to see how the algorithm performs in "harder" spaces than simply random ones.

---

> ### Author Rebuttal · Authors · 2023-08-09
>
> We thank the reviewer for their constructive feedback and valuable comments. We will revise the citations, notations, and definitions based on your suggestions. Below we answer your questions.
>
> - _How important is it to the overall community?_ Understanding how CFR+ works well in game solving is an important open question in the community. Because of the widespread use of these algorithms (for one, it was used in recent poker AI milestones, where poker AIs beat human poker players [1,2,3,4]), it has become an important open problem to explain their strong empirical behaviors, e.g. as mentioned in [5].
>     Apart from the regret guarantees in $O(\sqrt{T})$, prior to our work virtually nothing was known that could provide a sound theoretical argument explaining RM+ and CFR+.
>     Although we don't fully answer this question, in this paper we point out that (in)stability plays a crucial role in their empirical performances. Lacking stability can ruin CFR+ in some simple examples.
>     On the other hand, from the theoretical perspective, we are the first to show fast convergence of RM-based algorithms by stabilizing the algorithms. This not only greatly shrinks the discrepancy between theory (usually OMD-based) and practice (usually RM-based), but also sets a cornerstone for developing better and more robust variants of CFR+.
>
> - _Different threshold for $\|R_1\|$:_ Changing the threshold for $\| R \|_{1}$ has an impact on the Lipschitz constant of the function $g$ as defined in Proposition 1, line 159 of our submission. In particular, in our algorithms, there are 2 tunable parameters: the threshold you mentioned and the step size $\eta$. It suffices to fix the threshold and tune $\eta$ only. The theorems still hold when changing the threshold to another constant $C>0$ by adjusting $\eta$ accordingly.  We will provide more details on this in the final version of our paper.
> - _Clairvoyant CFR doesn't seem to perform as well with EFG as it does with matrix games:_ You are correct that Clairvoyant CFR appears to have weaker performances (Figure 3) than its counterpart ExRM+ on matrix games (Figure 2), as we also point out in the paper.
>     We do not currently have an explanation for this. However, note that it is consistent with some of the experiences for some of the other $O(1/T)$ convergence-rate methods such as e.g. mirror prox and optimistic mirror descent/FTRL in the prior literature. For those algorithms, it has been observed that they can sometimes beat RM+ on matrix games, but it is much harder to beat CFR+ on EFGs. We view this as a general open problem regarding the "hard" structure of EFGs that makes it more difficult to attain fast convergence rates than for matrix games. See e.g. [6,7] below for past observations of this phenomenon.
> - _Converting into strategic game equivalent:_ The equivalent strategic-form representation of an EFG is exponentially large in the EFG size, and thus not practical for most of the games that we consider (see appendix K.2 line 696 for the description of the EFGs used in the experiments).
> - _Did you try to create data that would be more "difficult" like maybe a space that is an annulus?_ We did not try this, but it is an interesting idea and we will attempt it for the final version. We remark that unlike extensive form games—for which established benchmark games exist—there are no clearly established normal-form benchmark games, and it seems common practice in the literature to experiment on random matrix games.
>
> We hope that we have answered your questions and addressed your concerns about the significance of our results.
> In light of our responses, we would like to invite you to reconsider the importance of our contributions.
>
> [1] M. Bowling, N. Burch, M. Johanson, and O. Tammelin. Heads-up limit hold’em poker is solved. Science,
>     347(6218):145–149, 2015.
>
> [2] N. Brown and T. Sandholm. Superhuman AI for heads-up no-limit poker: Libratus beats top professionals.
>     Science, 359(6374):418–424, 2018.
>
> [3] N. Brown and T. Sandholm. Superhuman AI for multiplayer poker. Science, 365(6456):885–890, 2019
>
> [4] M. Moravcık, M. Schmid, N. Burch, V. Lisy, D. Morrill, N. Bard, T. Davis, K. Waugh, M. Johanson, and M. Bowling. Deepstack: Expert-level artificial intelligence in heads-up no-limit poker. Science, 356(6337):508– 513, 2017.
>
> [5] Farina, G., Kroer, C., and Sandholm, T. (2021, May). Faster game solving via predictive Blackwell approachability: Connecting regret matching and mirror descent. In Proceedings of the AAAI Conference on Artificial Intelligence (Vol. 35, No. 6, pp. 5363-5371).
>
> [6] Gao, Yuan, Christian Kroer, and Donald Goldfarb. "Increasing iterate averaging for solving saddle-point problems." Proceedings of the AAAI Conference on Artificial Intelligence. Vol. 35. No. 9. 2021.
>
> [7] Farina, Gabriele, Christian Kroer, and Tuomas Sandholm. "Optimistic regret minimization for extensive-form games via dilated distance-generating functions." Advances in neural information processing systems 32 (2019).

---

> > ### Comment · Reviewer_GXrh · 2023-08-18
> > **I have read the author(s)'s rebuttal comments**
> >
> > Thank you for your comments. I have read them and I have changed my review accordingly.

---

### Official Review · Reviewer_NRiL · 2023-07-07

**Soundness:** 3 good
**Presentation:** 2 fair
**Contribution:** 2 fair
**Rating:** 6
**Confidence:** 3

**Summary:**

The paper studies variants of the RM+ algorithm for learning in games. They show that RM+ does not satisfy stability, which is an important property for proving good regret bounds, if all players play according to the same algorithm. The paper "fixes" the instability of RM+ by considering a restarting and chopping variant which stabilise RM+ and then prove O(T^{1/4}) and O(1) individual and social regret bounds respectively.


**Strengths:**

For the most part paper is well written and easy to follow.
The paper points to the importance of stability for proving sub-sqrt regrets bounds, which I find interesting.


**Weaknesses:**

In my opinion Sec 5 too dense and thus it lacks clarity. I would suggest to put only the main technical result there , or to give a skimmed version, in which the more technical assumptions are deferred to the appendix. I think that 4 statements with little discussion, cannot convey clarity.

**Questions:**

1. It seems that for the restarting trick you cannot prove O(1) social regret. Is this because restarting may happen at different times for different players? If so I think you should include a larger discussion on this. Moreover in the abstract it seems like both restarting and chopping would give O(1) social regret. I think you should state explicitly in the abstract that only holds for the chopping variant.
2. Do you conjecture that O(1) social is only hard to prove for the restarting method or that is not reached?

**Limitations:**

yes

---

> ### Author Rebuttal · Authors · 2023-08-09
>
> We thank you for your time reviewing the paper. We now answer your question and remarks on the manuscript.
>
> - Clarity: in the final version we will use the additional page to provide more details on the results in Section 5 (Conceptual RM+/ExRM+). This will improve the overall clarity of the paper.
> - Social regret for restarting (theory): we believe that it is hard to prove that restarting achieves $O(1)$ social regret; this is precisely for the reason that you give: restarting may happen asynchronously across players. We are not ready to make a conjecture about whether this is true or not.
>     We will provide more details on this in the revised version. We will also clarify the abstract.
> - Social regret for restarting (experiments): in our numerical experiments, we observed that restarting yields $O(1)$ social regret on the hard $3 \times 3$ matrix game instance (see Figure 7 in Appendix K). The numerical performances from Figure 2 also suggest that this is the case for random matrix game instances. However we did not search extensively over matrix game instances to try to disprove this statement, so we do not make any claims on this matter in the paper.
>
> Overall, we would like to emphasize that our paper is the first to provide some explanations on the very strong performances of RM+ and CFR+, which usually converge much faster than their $O(\sqrt{T})$ theoretical guarantees and routinely outperform theoretically stronger algorithms. Because of the widespread use of these algorithms (it was used in recent poker AI milestones, where poker AIs beat human poker players [1,2,3,4]), it has become an important open problem to explain their strong empirical behaviors, e.g. as mentioned in [5]. Apart from the regret guarantees in $O(\sqrt{T})$, prior to our work virtually nothing was known that could provide a sound theoretical argument explaining RM+ and CFR+. In light of this, we would like to invite you to reconsider the importance of our contributions, which identify the instability of RM+ as a key issue for proving stronger convergence rates, and to which we propose two practical solutions, restarting and clipping, leading to state-of-the-art convergence rates and strong practical performances for matrix games.
>
> [1] M. Bowling, N. Burch, M. Johanson, and O. Tammelin. Heads-up limit hold’em poker is solved. Science,
> 347(6218):145–149, 2015.
>
> [2] N. Brown and T. Sandholm. Superhuman AI for heads-up no-limit poker: Libratus beats top professionals.
> Science, 359(6374):418–424, 2018.
>
> [3] N. Brown and T. Sandholm. Superhuman AI for multiplayer poker. Science, 365(6456):885–890, 2019
>
> [4] M. Moravcık, M. Schmid, N. Burch, V. Lisy, D. Morrill, N. Bard, T. Davis, K. Waugh, M. Johanson, and M. Bowling. Deepstack: Expert-level artificial intelligence in heads-up no-limit poker. Science, 356(6337):508– 513, 2017.
>
> [5] Farina, G., Kroer, C., and Sandholm, T. (2021, May). Faster game solving via predictive Blackwell approachability: Connecting regret matching and mirror descent. In Proceedings of the AAAI Conference on Artificial Intelligence (Vol. 35, No. 6, pp. 5363-5371).

---

> > ### Comment · Reviewer_NRiL · 2023-08-12
> >
> > Thank you for addressing my concerns! I'll raise my score accordingly

---

### Official Review · Reviewer_jyGc · 2023-07-10

**Soundness:** 3 good
**Presentation:** 3 good
**Contribution:** 3 good
**Rating:** 7
**Confidence:** 3

**Summary:**

This paper studies the RM+ algorithm and its variants in the context of learning in games.
- They first show that RM+ and predictive RM+ can be unstable in certain environments. Although the unstable player benefits from instability, the regret of other players may blow up because they are forced to make predictions in an unstable environment.
- Then they provide several approaches to stabilize RM+.
	- The first approach, called **stable predictve RM+**, requires each player to (asynchronously) restart whenever $\mathbf{R}^t_i$ becomes small. Although this stabilizes the algorithm and guarantees $O(T^{1/4})$ regret for each player, it's hard to bound the social regret by $O(1)$ .
	- The second approach is called **smooth predictive RM+**. It chops off the area of decision space that is too close to the origin by adding projection steps. Besides individual regret bounds, this approach also guarantees $O(1)$ social regret.
	- They also propose **Conceptual RM+** (and the one fixed-point approximation of it) that achieves $O(1)$ individual regret and **Extragradient RM+** that achieves $O(1)$ social regret.
- The authors conduct experiments on matrix games and extensive-form games.

**Strengths:**

- The paper contributes to the theoretical understanding of the RM+ and its variants. It is the first one to show the instability of RM+ and predictive RM+ by concretely constructing a hard case that exhibits the $T^{-0.5}$ convergence rate.
- The paper proposes several variants of RM+ that not only have appealing theoretical guarantees on both individual regret and social regret, but also perform well in experiments.
- The paper is very well-written. The authors clearly explain the intuition behind the theoretical results.

**Weaknesses:**

- In section 3, the authors construct a hard case for RM+ and predictive RM+ where the empirical convergence rate are on the order of $T^{-0.5}$. However, the residue errors of the linear fit in Figure 1 are quite large. Providing a theoretical proof that the asymptotic convergence rate is indeed $T^{-0.5}$ would strengthen the result.
- Given the connection between RM+ and OMD in [12] and the fast convergence properties of OMD in games, the performance guarantee for the stabilized variants of RM+ do not seem surprising. Nonetheless, as mentioned by the authors, RM+-based algorithms are not inherently stable, so they need extra considerations.
- The experiments show the strong performances of the proposed algorithms compared to RM+ and PRM+. It would be better if the authors also compared their performance with the optimistic/predictive FTRL/OMD algorithms that enjoy fast convergence rates in theory.

**Questions:**

In the proof of Theorem 3.1, the authors constructed an example where the precision of the losses increases with $T$ (the minimum value is exponentially small), a scenario that may not practically appear.
- Is this phenomenon of increasing precision the fundamental reason of instability, or is it merely an artifact of the proof?
- Does this same phenomenon appear in the subsequent experiments involving the $3\times3$ matrix game?
- If a bounded precision assumption were introduced, could this potentially stabilize RM+ or PRM+?

**Limitations:**

Yes

---

> ### Author Rebuttal · Authors · 2023-08-09
>
> We thank you for your time reviewing the paper.
> - We have run additional numerical experiments to compare Optimistic OMD (OOMD) and Optimistic FTRL (OFTRL) with ExRM+, Stable and Smooth PRM+. We ran these algorithms for the $3 \times 3$ matrix game instance and for  $10$ random matrix game instances of size $15 \times 15$, with a step size of $\eta = 0.1$ for all algorithms. We present the empirical results in Figure 1 in the PDF attached to our response common to all reviewers. We found that ExRM+, Smooth and Stable PRM+ perform on par with Optimistc FTRL and Optimistic OMD, two methods that achieve the theoretical state-of-the-art guarantees for solving two-player zero-sum games.
>
> Question about the diminishing norm of the loss: We believe that this is simply an artifact of the proof, as we do not observe this phenomenon for our $3 \times 3$ matrix game instance.
>
> - We have verified this with additional numerical experiments, by computing the $\ell_{2}$-norm of the losses faced by each player at every iteration and the minimum absolute value of the coefficients of the losses faced by each player at every iteration. We refer to Figure 2 in the attached PDF document (in the responses common to all reviewers). As you can note, we found that these two quantities become in the order of $10^{-1}$ after $10^3$ iterations but do not converge to $0$. Therefore we do not believe that a "bounded precision assumption" would stabilize RM+ or PRM+.
>
> Finally, you say "_Given the connection between RM+ and OMD in [12] and the fast convergence properties of OMD in games, the performance guarantee for the stabilized variants of RM+ do not seem surprising._"
>
> - We disagree with this framing of our paper. While it is true that [12] shows a connection to OMD, it has been an open problem since [12] whether (predictive) RM+ can achieve a fast convergence as with optimistic OMD. Whether our result is "surprising" or not is of course debatable (and perhaps not a good way to evaluate it one way or the other), but we do believe that this was a significant problem that had been open for a few years. Reducing it to "not surprising due to OMD" is too dismissive, in our view.
>     Moreover note that, in a sense, we resolve the PRM+ open problem in the \emph{negative}, since our counterexamples show that in fact PRM+ does not achieve a faster rate. This is in spite of the fact that PRM+ is also equivalent to optimistic OMD (on a different feasible set than our stabilized variants). Only through our stabilizing ideas is a faster rate possible.

---

> > ### Comment · Reviewer_jyGc · 2023-08-15
> >
> > Thank you for the response and for running the additional experiments. My concerns have been adequately addressed, and I will retain my original score of 7.

---

### Official Review · Reviewer_zLtq · 2023-07-10

**Soundness:** 4 excellent
**Presentation:** 3 good
**Contribution:** 3 good
**Rating:** 8
**Confidence:** 3

**Summary:**

The paper focuses on investigating the stability of the Regret Matching+ based algorithm. It demonstrates that both RM+ and predictive RM+ algorithms exhibit instability, leading to significant regret for other players within a game setting. To address this issue, the authors propose two methods: restarting and chopping off. By incorporating these methods, the authors introduce several variants of RM+-based algorithms, including stable/smooth Predictive RM+, conceptual RM+, and Extragradient RM+. The paper also provides theoretical regret bounds for these proposed algorithms, further establishing their efficacy in mitigating instability and reducing regret.

**Strengths:**

One of the key contributions of this paper is the theoretical speedup provided for the RM+ based algorithm, which is crucial for achieving fast convergence in games. By offering improved convergence guarantees, the proposed advancements in the algorithm can significantly enhance its practical utility.

The writing in the paper is well-organized and clear, effectively conveying the technical details. The strength of the paper's technical aspects further adds to its overall quality, making it more accessible and understandable for readers.

The comprehensive experimental results presented in the paper further enhance its credibility. Notably, the authors also consider the application of the algorithm in extensive form games, which is important for real-world scenarios. This consideration highlights the practical relevance and versatility of the proposed techniques.

Overall, the paper provides valuable theoretical insights, demonstrates technical proficiency, and showcases the practical applicability of the proposed algorithm, making it a strong contribution to the field.

**Weaknesses:**

Just a small point about presentation. It would be beneficial for the authors to consider organizing the algorithms discussed in the paper by creating a table. This table can help provide a clear overview of the different algorithms, highlighting their similarities and differences.

**Questions:**

The thoerem need $\eta$ to be small. However, in the matrix game experiment, $\eta$ is chose as $10$. Is $\eta = 10$ too big in this setting?

**Limitations:**

The authors have adequately addressed the limitations.

---

> ### Author Rebuttal · Authors · 2023-08-09
>
> We thank you for your time reviewing the paper. In the final version of the paper, we will add a table to summarize the different regret guarantees for the algorithms introduced in this paper. Regarding your question: you are right that the stepsizes we use in the experiments are generally larger than what the theory requires. This is a commonly-found phenomenon, that the theoretical stepsizes are too conservative, see e.g. the related literature in [10,25]. We will make sure to emphasize this more in the final version.

---

> > ### Comment · Reviewer_zLtq · 2023-08-15
> >
> > Thank you for your response!

---

### Author Rebuttal · Authors · 2023-08-09

We thank all the reviewers. We responded to each of your questions in individual comments. Here we upload a PDF of two figures to support those responses. Specifically, in the PDF, we plotted:

- Figure 1: Comparison of Optimistic OMD, Optimistic FTRL, ExRM+, Stable and Smooth PRM+ on random matrix instances and our hard $3 \times 3$ matrix game instance.

- Figure 2: Minimum absolute value of the losses of each player and $\ell_{2}$-norm for the loss of each player, for the $3 \times 3$ hard matrix game instance.

---

### Decision · Program_Chairs · 2023-09-21

**Decision:**

Accept (spotlight)

**Comment:**

This paper focuses on regret matching (a standard no-regret algorithms) in games where it might become unstable. The authors nicely prove that fix a few fixes, those properties can be recovered.

All the reviewers, and myself, enjoyed that paper !